# Performance and emission optimization of a CRDI engine in RCCI mode using hydrogen enriched biodiesel through grey relational analysis approach

**Aashish Divya Shinil Kumar, Vishakha Vijayashankar Hebballi, Jibitesh Kumar Panda** ⓘ*

Department of Mechatronics, Manipal Institute of Technology, Manipal Academy of Higher Education, Manipal, Karnataka, India

* jibitesh.panda@manipal.edu

## Abstract

This study investigates the performance and emission optimization of a Common Rail Direct Injection (CRDI) diesel engine operated in Reactivity Controlled Compression Ignition (RCCI) mode using alternative fuel strategies. Experiments were performed on a single-cylinder, four-stroke, water-cooled CRDI engine operating at a constant speed of 1500 rpm under 25%, 50%, 75%, and 100% load conditions, corresponding to a brake mean effective pressure (BMEP) range of 0.4–1.6 MPa. The test environment was maintained at an ambient temperature of $29 \pm 2$ °C, relative humidity of $68 \pm 5\%$, and atmospheric pressure of 1.01 bar. The intake manifold pressure was regulated at $1.0 \pm 0.02$ bar, with coolant and lubricating oil temperatures stabilized at $80 \pm 2$ °C and $85 \pm 3$ °C, respectively, to ensure consistent thermal boundary conditions. Four fuel configurations were evaluated: neat diesel (D100), CNG+D100, $H_2$+D100, and hydrogen-enriched biodiesel (B20+$H_2$). Hydrogen and CNG were inducted through the intake manifold at flow rates of 2, 4, 6, and 8 L/min, while diesel and biodiesel served as pilot fuels injected at a fixed pressure of 500 bar. Engine performance and emission characteristics were analysed in terms of Brake Thermal Efficiency (BTE), Brake Specific Fuel Consumption (BSFC), and exhaust emissions (CO, HC, $NO_x$, and smoke opacity). Grey Relational Analysis (GRA) was applied for multi-response optimization, identifying B20+$H_2$ at 2 L/min as the most favourable condition, offering higher BTE, lower BSFC, and reduced emissions. The study highlights the dual advantage of hydrogen-enriched biodiesel in achieving a cleaner and more efficient RCCI operation. The work aligns with UN SDG-7 and SDG-13 by advancing a cleaner, low-carbon dual-fuel pathway through hydrogen-enriched biodiesel combustion in modern engines.

**Data availability statement:** All relevant data are available via Figshare using the following DOI: 10.6084/m9.figshare.30596354.

**Funding:** The author(s) received no specific funding for this work.

**Competing interests:** The authors declare that they have no competing interests.

## 1. Introduction

Recent diesel engine advancements, especially CRDI technology [1–4], address emission norms and fuel efficiency demands, significantly reducing fuel consumption and particulates compared to traditional systems [3–9]. However, the same technologies often result in elevated NOx emissions due to higher in-cylinder temperatures under lean and homogeneous combustion conditions [10]. Exhaust Gas Recirculation and pilot injection strategies have been investigated extensively to mitigate these emissions while maintaining efficiency [11–18]. The global emphasis on reducing fossil fuel dependency and addressing air pollution challenges has highlighted the need for alternative fuels. Among these, compressed natural gas has drawn attention due to its lower emissions profile and near-term sustainability [17–24]. Hydrogen, however, offers a promising long-term solution with zero carbon emissions at the point of use and high energy density, positioning itself as a key candidate in a future low-carbon economy [24–32]. In this context, the trade-off between performance (brake thermal efficiency and brake specific fuel consumption) and emissions (NOx, UHC, smoke, and CO) becomes increasingly critical for modern engines [29–37]. Achieving an optimal balance requires advanced control strategies and combustion concepts such as Reactivity Controlled Compression Ignition (RCCI), which uses fuels of different reactivities to promote cleaner combustion [17–21]. The present study focuses on a Kirloskar single-cylinder, four-stroke, water-cooled Variable Compression Ratio (VCR) CRDI diesel engine operating in RCCI mode. Four fuel configurations D100, compressed natural gas, hydrogen, and B20 blended with hydrogen were tested at a fixed maximum load of 12 kg and constant injection pressure of 500 bar, while varying the flow rates (2, 4, 6, and 8). The study used Grey Relational Analysis [19,20] to evaluate CO, NOx, UHC, smoke, BSFC, and BTE, identifying RCCI fuel blends as efficient, cleaner alternatives [31–39], despite challenges with CNG, hydrogen, and biodiesel emissions. Prior studies have predominantly focused on single-response optimizations without fully integrating multi-objective approaches like Grey Relational Analysis to identify ideal operational strategies in dual-fuel RCCI configurations [40,41].

Existing optimization studies in compression ignition and RCCI engines have generally applied either the Taguchi–Grey Relational Analysis (GRA) method or Artificial Neural Network (ANN)-based models to identify optimal parameters such as injection timing, pressure, and blending ratio [29,31,36,41,42]. However, these methods were often limited to *single-fuel systems* or *simulation-based datasets* and did not simultaneously address hydrogen enrichment, biodiesel reactivity, and dual-fuel control variables in a real-time CRDI engine. The present work introduces several novel contributions given below.

- A hybrid Taguchi–GRA framework experimentally implemented for hydrogen-enriched biodiesel RCCI operation, enabling direct multi-response optimization of both efficiency and emission indices.

- Evaluation of hydrogen induction rates (2–8 LPM) and Palm Kernel Methyl Ester (B20) in a 500 bar CRDI-RCCI configuration, establishing a new operational benchmark for dual-fuel engines.

- Integration of ANOVA-based statistical validation quantifies factor influence (Fuel Strategy ≈ 51.4%), improving interpretability compared with black-box ANN models.

- Inclusion of a Life-Cycle Assessment (LCA) linking laboratory optimization with real-world carbon reduction potential.

These combined elements make this study a comprehensive experimental–analytical–sustainability framework that surpasses prior Taguchi–GRA or ANN-only optimizations, contributing both methodological depth and environmental relevance to RCCI research.

Unlike earlier hydrogen–biodiesel dual-fuel investigations that examined single-parameter variations or limited performance–emission responses, the present work uniquely integrates Taguchi–Grey Relational Analysis (GRA) to achieve multi-response optimization under RCCI operation. The study employs a CRDI engine platform operating at 500 bar injection pressure and systematically evaluates hydrogen enrichment rates (2, 4, 6, and 8 LPM) with a Palm Kernel Methyl Ester (PKME-based B20) biodiesel blend as the pilot fuel. This combination enables a holistic evaluation of performance and emission trade-offs, establishing the optimal hydrogen flow rate of 2 LPM for balanced BTE and low-emission operation. Such integration of advanced optimization with renewable hydrogen-biodiesel combustion under RCCI conditions has not been previously reported, underscoring the originality and applied relevance of the present study.

The optimization of internal combustion engine parameters is inherently complex due to the conflicting behavior of performance and emission characteristics. For instance, improving brake thermal efficiency often results in an increase in NOx emissions, while lowering smoke and carbon monoxide may compromise specific fuel consumption. Traditional single-objective optimization techniques fail to capture these trade-offs effectively, limiting their applicability in dual-fuel and hydrogen-assisted RCCI operations. Moreover, experimental trial-and-error approaches are time-intensive, resource-heavy, and unable to reveal the underlying interdependencies between control factors. In this context, multi-objective optimization tools offer a systematic pathway to balance these responses simultaneously. Among the available techniques, Grey Relational Analysis (GRA) has emerged as a robust and computationally efficient method, capable of handling non-linear, incomplete, or uncertain information that often characterizes engine performance data [40–46].

GRA converts multiple performance and emission responses into a single relational grade through normalization and grey relational coefficients, enabling a holistic evaluation of experimental outcomes. Unlike conventional regression or machine-learning-based approaches, which require large datasets and intensive computation, GRA provides reliable insights even with limited experimental runs, making it highly suitable for experimental engine studies. Additionally, when combined with Taguchi's Design of Experiment, GRA reduces the number of tests required while ensuring statistical soundness in parameter selection [45–47]. This integration not only enhances the accuracy of identifying optimal fuel strategies but also aligns with sustainability objectives by minimizing experimental effort and fuel usage. Therefore, in the present study, GRA is employed to identify the optimal combination of fuel type, injection parameters, and hydrogen enrichment rate, ensuring a balanced improvement in thermal efficiency and emission control. This justification highlights the appropriateness of GRA in addressing the multi-response optimization challenge inherent in RCCI research.

## 1.1. Based on the review, the following research gaps have been identified

- Limited experimental studies exist on the performance of hydrogen-enriched biodiesel blends in RCCI mode using a CRDI engine.

- Few investigations comprehensively evaluate multiple performance and emission parameters simultaneously under varying flow rates.

- Optimization studies using combined Taguchi and GRA techniques remain underexplored for dual-fuel RCCI operation.

## 1.2. Objectives of the study

- **Experimental Evaluation** – To systematically investigate the performance and emission behavior of a CRDI diesel engine operating in RCCI mode using four fuel strategies (D100, CNG, $H_2$, and B20 + $H_2$) at multiple flow conditions.

- **Optimization through GRA–Taguchi Approach** – To apply a combined Design of Experiments (DoE) and Grey Relational Analysis framework for optimizing key response parameters, including Brake Specific Fuel Consumption (BSFC), Brake Thermal Efficiency (BTE), NOx, CO, UHC, and smoke emissions.

- **Validation of Findings** – To conduct confirmation experiments that validate the optimized fuel-flow configuration and demonstrate its effectiveness in achieving a favorable performance emission trade-off.

## 2. Pilot fuels and their properties

In a Reactivity Controlled Compression Ignition (RCCI) engine, the careful selection and combination of fuels determine the overall combustion characteristics, thermal efficiency, and emission outcomes. The RCCI principle is based on blending fuels with different reactivity levels so that ignition timing and combustion phasing can be optimized for improved performance and reduced pollutant formation. By adjusting the share of high-reactivity and low-reactivity fuels, it is possible to achieve a more controlled combustion process, thereby addressing two of the major challenges faced by conventional compression ignition engines high nitrogen oxide (NOx) formation and excessive soot emissions. In the present study, four distinct fuel strategies were considered to evaluate their suitability for RCCI operation: neat diesel (D100), compressed natural gas with pilot diesel (CNG), hydrogen–diesel (D100 + $H_2$), and biodiesel–hydrogen (B20 + $H_2$). Each of these fuel combinations was deliberately selected for its unique chemical and combustion properties, enabling a comprehensive assessment of both performance and emissions. Neat diesel (D100) served as the baseline fuel due to its widespread usage and well-established ignition quality. Diesel has a high cetane number, ensuring reliable autoignition, but it is also associated with elevated soot emissions under conventional operation. In dual-fuel RCCI mode, however, diesel plays a critical role as the high-reactivity component that governs ignition timing and stabilizes combustion. Compressed natural gas (CNG) represents a low-reactivity fuel with a high methane content, providing slower ignition and extended combustion duration. When used in dual-fuel mode with pilot diesel, CNG contributes to cleaner combustion, particularly through reduced particulate matter and carbon monoxide emissions. However, due to its relatively high autoignition temperature, it requires diesel assistance to initiate combustion.

Hydrogen diesel (D100 + $H_2$) combines the stable ignition quality of diesel with the fast flame propagation and clean-burning nature of hydrogen. Hydrogen has an exceptionally high diffusivity and zero carbon content, which not only promotes more complete combustion but also minimizes carbon-based emissions such as CO, HC, and smoke. Nonetheless, the high flame temperature of hydrogen can lead to an increase in NOx, requiring careful optimization of flow rates and operating parameters. Finally, the biodiesel hydrogen (B20 + $H_2$) strategy introduces a renewable, oxygenated fuel in combination with hydrogen. Biodiesel offers additional oxygen within its molecular structure, supporting complete combustion, while hydrogen enrichment enhances flame speed and reduces unburned hydrocarbons. This combination aims to provide a balanced approach by lowering CO, HC, and smoke emissions while maintaining high thermal efficiency. Thus, the selected fuel strategies highlight contrasting reactivity levels, enabling the RCCI engine to be systematically studied for performance emission trade-offs and optimized through advanced techniques such as Taguchi–Grey Relational Analysis.

## 2.1. Role of individual fuels

- **Diesel (D100):** Acts as a baseline fuel due to its high cetane number, short ignition delay, and reliable combustion. However, it tends to produce higher particulate matter (PM), CO, and smoke under conventional operation.

- **Compressed Natural Gas (CNG):** With high octane and low cetane numbers, CNG requires pilot diesel for ignition. It burns more cleanly than diesel, reducing $CO_2$ and smoke but showing limitations in terms of lower thermal efficiency at low loads.

- **Hydrogen ($H_2$):** A carbon-free fuel with exceptionally high flame speed, wide flammability limits, and low ignition energy. Hydrogen enhances mixture homogeneity and accelerates combustion, reducing CO, UHC, and smoke. However, it can increase peak in-cylinder temperature, leading to higher NOx.

- **Biodiesel (B20 blend):** The oxygenated nature of biodiesel promotes more complete combustion, reducing CO and smoke. However, biodiesel alone may increase NOx emissions.

## 2.2. Synergistic roles in RCCI setup

- **D100+$H_2$:** Diesel provides ignition, while hydrogen improves combustion completeness and lowers CO/HC emissions.

- **B20+$H_2$:** Combines the renewable, oxygenated character of biodiesel with hydrogen's high diffusivity, enabling significant smoke and CO reductions while maintaining efficiency.

- **CNG+Diesel:** Leverages diesel for ignition stability while utilizing CNG's low carbon intensity to reduce overall emissions.

## 2.3. Key benefits of fuel combinations

- Enhanced thermal efficiency due to hydrogen's rapid flame speed.

- Reduced CO and smoke through biodiesel's oxygenated nature.

- Controlled ignition via diesel pilot injection ensuring RCCI stability.

- Balanced trade-off between performance and emissions when fuels are strategically combined.

Thus, the selected fuel strategies exploit the contrasting properties of diesel, biodiesel, hydrogen, and CNG to optimize RCCI combustion. Among these, the B20+$H_2$ combination emerges as the most effective, delivering lower emissions without compromising efficiency.

In this study, the biodiesel used was Palm Kernel Methyl Ester (PKME), prepared through the transesterification of palm kernel oil. The B20 blend was obtained by mixing 20% PKME with 80% conventional diesel. PKME was selected owing to its relatively high oxygen content (~11%), moderate calorific value (37–39 MJ/kg), and suitable cetane number (45–55), which collectively support improved combustion efficiency and reduced carbonaceous emissions when compared to neat diesel. The presence of oxygen in PKME also promotes more complete oxidation, thereby lowering CO and smoke formation under RCCI operation.

The Palm Kernel Methyl Ester (PKME) used in this study was tested in the laboratory prior to blending to confirm its compliance with biodiesel standards. Standard test procedures were followed: calorific value (ASTM D240), density (ASTM D1298), kinematic viscosity (ASTM D445), cetane number (ASTM D613), and flash point (ASTM D93). The measured properties are summarized in Table 1(a). These values are consistent with reported PKME properties in the literature [2,20] and validate its suitability for dual-fuel RCCI operation. Table 1 represents the chemical properties of the current fuel for the experiments.

Diesel and biodiesel fuels are typically defined by their cetane number (CN), which reflects ignition quality. In contrast, gaseous fuels such as CNG and hydrogen are low-reactivity fuels and do not possess meaningful CN values. Instead, they are characterized by high octane numbers (ON), which represent their strong resistance to auto-ignition. In this study, the ON values were initially misrepresented as CN; this has now been corrected.

**Table 1. Chemical properties of the fuels.**

| Fuel Type | Calorific Value (MJ/kg) | Density (kg/m³) | Cetane Number | Viscosity (cSt at 40°C) | Carbon (%) | $H_2$ (%) | $O_2$(%) | Flash Point (°C) | Autoignition Temp (°C) |
|---|---|---|---|---|---|---|---|---|---|
| Diesel | 42–45 | ~830 | 50–55 | 2.8–3.2 | ~86 | ~13 | ~0 | 60–65 | 210–260 |
| CNG | ~50 (per kg) | 0.717 (STP) | ~130 | – | | ~75 | ~25 | 0 | – | ~540 |
| $H_2$ | ~120 (per kg) | 0.0899 (STP) | >130 | – | 0 | 100 | 0 | – | ~585 |
| PKME | 37.8 | 860 | 49 | 4.5 | ~77 | ~12 | ~11 | 163 | 250–300 |
| B20 | 39.8 | 842 | 51 | 3.6 | ~83 | ~13 | ~4 | 82 | 230–270 |

**\*B20 values were measured experimentally and fall within ASTM D7467 standards for biodiesel blends.**

## 3. Instrumentation

The experimental investigations were conducted on a single-cylinder, four-stroke, water-cooled Variable Compression Ratio (VCR) diesel engine developed by Kirloskar, which was suitably modified for dual-fuel operation under Reactivity Controlled Compression Ignition (RCCI) mode. The baseline engine was equipped with a modern Common Rail Direct Injection (CRDI) system, which consisted of a high-pressure fuel pump, common rail, solenoid-controlled injector, and an advanced Electronic Injection Controller (EIC) to govern the timing and quantity of diesel injection with high accuracy. The CRDI system was capable of handling injection pressures up to 600 bar, thereby ensuring superior atomization of diesel and biodiesel fuels, which directly enhanced combustion efficiency and provided better control over in-cylinder processes [4,7,8]. The detailed specification of the CRDI engine is provided in Table 1(B), while Table 2 outlines the configuration of the EIC. To accommodate dual-fuel operation, the engine was retrofitted with a dedicated gaseous fuel supply system designed to handle both hydrogen and compressed natural gas (CNG). In this arrangement, hydrogen or CNG was inducted into the intake manifold using a calibrated manifold injection kit, which ensured uniform mixing with the intake air prior to combustion. The manifold injection kit was supported by a programmable Open Loop Electronic Control Unit (OPECU), which controlled injection timing and pulse duration for the gaseous fuels, allowing precise regulation of flow rates at 2, 4, 6, and 8 L/min as required in the test matrix [24,25]. The experimental setup thus ensured that diesel or biodiesel (B20) served as the high-reactivity pilot fuel injected directly through the CRDI system, while hydrogen or CNG functioned as the low-reactivity fuel inducted via the manifold, thereby facilitating RCCI combustion with improved mixing and reduced ignition delay. Fig 1 depicts the complete setup for experimentation, highlighting the CRDI system, secondary gaseous fuel circuit, dynamometer, and emission analyzers. For load application and performance measurement, an eddy current dynamometer coupled with strain gauge and load cell-based measurement units was used, providing accurate quantification of torque and brake power at various operating conditions [23]. Exhaust emissions were continuously measured using a five-gas analyzer (for CO, $CO_2$, HC, NOx, and $O_2$ concentrations), while particulate emissions and soot density were evaluated using an AVL 415S smoke meter, ensuring comprehensive monitoring of engine-out emissions [26].

During dual-fuel operation, the intake manifold pressure was continuously monitored using a Kistler piezo-resistive transducer (range: 0–2 bar, accuracy: ±0.25%) connected near the manifold junction. The sensor output was logged through a National Instruments (NI) DAQ system interfaced with LabVIEW to detect any pressure fluctuation beyond ±0.02 bar. Simultaneously, the equivalence ratio (φ) was calculated in real time based on the measured air mass flow (via orifice meter and U-tube manometer) and gaseous fuel flow rates (from calibrated MFCs). The average equivalence ratio was maintained at 0.85 ± 0.03 across all hydrogen and CNG trials, ensuring uniform air–fuel mixing and minimizing charge stratification. The stability of manifold pressure and equivalence ratio confirmed consistent combustion conditions during RCCI experiments, thus supporting reproducibility of the obtained performance and emission data.

Prior to each experimental run, the hydrogen and CNG supply lines were equipped with Bronkhorst-type mass flow controllers (MFCs) rated for 0–10 L/min and certified to ±1% full-scale accuracy. The MFCs were calibrated against a

**Table 2. Engine specification.**

| Parameter | Specifications |
|---|---|
| Make and type | Kirloskar, mono-cylinder, four stroke, CRDI engine |
| Stroke length | 110m |
| Power (kilowatts) | 3.5 (1500 rpm) |
| Dynamometer | Eddy current type with power and torque measurement |
| Make and load measurement | Eddy current dynamometer with strain gauge and load cell-based measurement |
| Type and cooling | Eddy current dynamometer; water-cooled engine system |

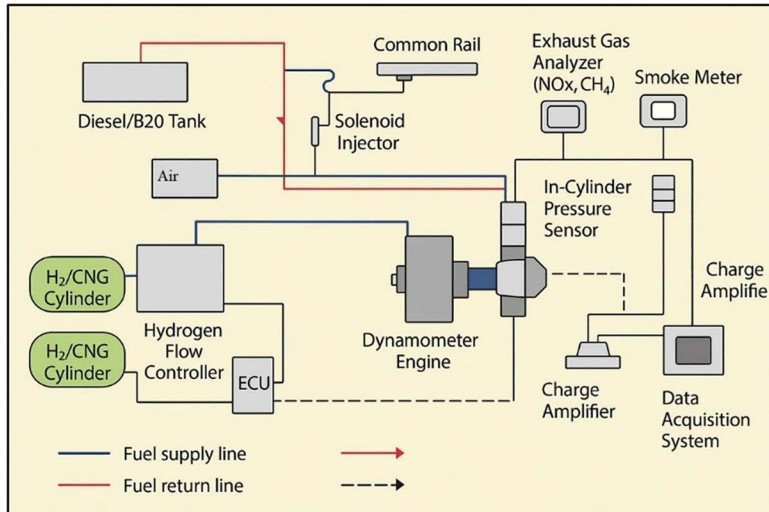

**Fig 1. Setup for experimentation.**

precision rotameter and soap-film flow meter to validate linear response over the entire operating range. Flow readings were cross-checked at 2, 4, 6, and 8 L/min under atmospheric conditions to ensure consistency between devices. The controllers were integrated with the Open-Loop Programmable Electronic Control Unit (OPECU), which generated constant-pulse signals for gaseous fuel induction. Before every test sequence, the calibration was re-verified, and flow-rate deviations were found to remain within ±0.05 L/min. This procedure guarantees reproducibility of hydrogen and CNG induction conditions across all experiments.

Additionally, hydrogen energy share during dual-fuel operation was calculated using established thermochemical relations [27], ensuring proper evaluation of its contribution to the combustion process. The integration of the CRDI system and gaseous fuel manifold created a robust platform for analyzing the combined effect of injection pressure, load, and flow rate on combustion, performance, and emissions. This structured instrumentation approach enabled a systematic assessment of hydrogen-enriched biodiesel blends (B20 + H$_2$) in comparison to D100, D100 + H$_2$, and CNG strategies, thereby providing accurate experimental data to validate the optimization framework applied later in the study [19,20,32]. Overall, the detailed instrumentation ensures reproducibility, reliability, and clarity in measuring performance parameters such as Brake Thermal Efficiency (BTE), Brake Specific Fuel Consumption (BSFC), and critical emission indicators, while maintaining consistency between the CRDI and manifold-based secondary fuel systems.

### 3.1. Instrumentation details and calibration protocols

The experimental setup employed a suite of laboratory-grade instruments for pressure, temperature, speed, and emission measurements. Cylinder pressure was measured using a Kistler 6052C piezoelectric transducer (range: 0–200 bar; accuracy: ±0.25% F.S.; response time: <1 µs) coupled with a Kistler 5011B charge amplifier. Crank-angle position was captured using a Kubler 8.5868 incremental encoder (720 pulses/rev; accuracy: ±0.1 °CA). Intake manifold pressure was monitored by a WIKA S-10 piezoresistive transducer (0–2 bar; accuracy: ±0.25% F.S.), and airflow was measured using an orifice flowmeter connected to an inclined manometer (range: 0–600 LPM; accuracy: ±1%).

Exhaust emissions were quantified using an AVL Digas 444 five-gas analyzer (CO: 0–10% vol, HC: 0–20,000 ppm, $CO_2$: 0–20%, NOx: 0–5000 ppm; response time: ≤1.5 s) and an AVL Smoke Meter (437C) for opacity (range: 0–100%; response: <2 s). All emission analyzers were subjected to zero and span calibration before each experimental session using certified calibration gases (CO: 1%, $CO_2$: 10%, NO: 3000 ppm, $CH_4$: 1000 ppm in $N_2$ balance) from *Chemtron Science Laboratories, India*. The gas lines were purged before each run to prevent contamination.

Temperature measurements for coolant, lubricating oil, and exhaust were performed using K-type thermocouples (range: 0–1200 °C; accuracy: ±1 °C; response: <0.5 s). Hydrogen and CNG mass flow controllers (Bronkhorst EL-FLOW series; 0–10 LPM; accuracy: ±1% F.S.) were recalibrated against a rotameter and soap-film meter at the start of each test day. Data acquisition was managed through a National Instruments (NI-USB 6210) system interfaced with LabVIEW 2018, ensuring synchronized high-speed recording of all engine parameters. The zero/span calibration procedure was repeated at the beginning and end of each test cycle, and deviations exceeding ±2% were corrected before proceeding. These rigorous calibration and validation steps ensured the accuracy and repeatability of all recorded measurements.

### 3.2. Hydrogen safety and laboratory compliance

All hydrogen-related experiments were performed under controlled laboratory conditions that complied with the Apex Innovation Pune, India Laboratory Safety Protocol and national guidelines (BIS IS 16046/ ISO 26142: Hydrogen Detection Systems). The hydrogen cylinders (99.99% purity, 150 bar) were located in a well-ventilated, explosion-proof enclosure fitted with automatic leak detectors (Honeywell Sense point XCL, 0–2% $H_2$ vol. range, response <3 s). Gas flow from the high-pressure regulator to the manifold was equipped with flashback arrestors, non-return valves, and pressure-relief vents. A flame-proof solenoid isolation valve automatically shut off supply if $H_2$ concentration exceeded 0.4% vol. or power was interrupted. The entire test cell was continuously ventilated (air exchange ≈ 12 times $h^{-1}$) and grounded to prevent static discharge.

Prior to daily testing, all joints were checked using a soap-solution and electronic leak-detection test; no measurable leakage (< $1 \times 10^{-4}$ mbar L $s^{-1}$) was observed. Operators wore anti-static clothing and eye protection, and ignition sources were restricted within a 5 m radius. The emergency response plan was reviewed by the Institutional Laboratory Safety Committee, and all personnel completed mandatory hydrogen-handling training. These measures ensured that hydrogen use remained well below lower explosive limits (LEL = 4% vol.), and all trials were conducted in full compliance with institutional and national laboratory safety standards.

## 4. Methodology

The methodology of this research was designed to systematically evaluate the performance and emission behaviour of a CRDI engine operating in RCCI mode under different dual-fuel strategies and subsequently optimize the outcomes using Grey Relational Analysis (GRA). The experimental setup consisted of a single-cylinder, four-stroke, water-cooled, Variable Compression Ratio (VCR) diesel engine manufactured by Kirloskar, retrofitted with a Common Rail Direct Injection (CRDI) system capable of varying injection pressures between 100–120 MPa and controlled through a Bosch Electronic Injection Controller (EIC), with details provided in Tables 2 and 3.

**Table 3. Specification of the EIC.**

| Specification | Details |
|---|---|
| Type | Common Rail Direct Injection (CRDI) System |
| Make | Bosch |
| Injection Pressure | 100–120 MPa |
| Number of Holes | 5 (symmetrically arranged) |
| Nozzle Diameter | 0.15 mm |
| Injection Angle | 120° |

The test rig was coupled to an eddy current dynamometer for applying variable load conditions, while a 5-gas analyser and AVL 415S smoke meter were employed to measure CO, $CO_2$, HC, NOx, $O_2$, and smoke opacity, ensuring precise quantification of both performance and emissions [23,26]. The experiments were conducted at a constant speed of 1500 rpm with four load variations (25%, 50%, 75%, and full load), where the baseline D100 operation was compared against dual-fuel strategies of CNG + D100, $H_2$ + D100, and $H_2$ + B20 with hydrogen and CNG introduced through a manifold injection kit controlled via an open-loop programmable electronic control unit (OPECU) [24,25]. Diesel injection was initiated at 5° BTDC for all cases, and hydrogen flow rates were maintained at 2, 4, 6, and 8 LPM. These combinations were selected to investigate combustion stability, brake thermal efficiency (BTE), brake specific fuel consumption (BSFC), unburned hydrocarbons (UHC), CO, NOx, and smoke emissions, which are critical indicators of RCCI engine performance [19,20].

In all test conditions, the Common-Rail Direct Injection (CRDI) system was operated at a constant rail pressure of 500 bar, ensuring uniform atomization and injection timing precision. The pilot start-of-injection (SOI) was fixed at –10 °CA before top dead centre (bTDC), followed by the main injection commencing at –3 °CA bTDC, giving an effective dwell period of 7 °CA between injections. The total injection duration varied between 0.85 ms and 1.05 ms depending on the load condition, governed automatically by the electronic injector controller. To monitor combustion phasing, in-cylinder pressure was recorded using a Kistler 6052C pressure transducer with crank-angle resolution of 0.1 °CA, and the data were analyzed through LabVIEW-based acquisition software. The CA50 (crank angle corresponding to 50% heat-release completion) and ignition delay (ID) were determined from the cumulative heat-release rate curve. Across all operating points, CA50 was maintained within 8–10 °CA aTDC, while ignition delay ranged from 6.5–8 °CA, ensuring consistent combustion phasing. These parameters were not externally controlled but continuously monitored to confirm repeatability between runs. The constant rail-pressure operation and controlled SOI-dwell configuration ensured stable reactivity-controlled combustion under all dual-fuel strategies.

The experimental phase was followed by the implementation of the Design of Experiments (DoE) framework employing a Taguchi L16 orthogonal array to minimize the number of trials while preserving statistical accuracy, with the input factors set as fuel flow rate, injection pressure, and fuel strategy, as described in Tables 3 and 4 [29,30]. Signal-to-Noise (S/N) ratios were calculated for each response variable, where "smaller-the-better" criteria were used for BSFC, NOx, UHC, CO, and smoke, while "larger-the-better" criteria were applied for BTE, and these values were further normalized for use in GRA [38,40].

All experiments were performed at a constant 1500 rpm, which corresponds to the rated speed of the Kirloskar CRDI engine used in this study. This speed ensures stable injection timing synchronization with the Electronic Injection Controller (EIC) and minimizes cyclic variation, especially in RCCI mode where two fuels of different reactivities are involved. Operating at 1500 rpm also aligns with typical agricultural and stationary diesel applications, providing a practical baseline for fuel–emission evaluation [7,8,23].

At higher engine speeds (e.g., 2000–2500 rpm), the reduced residence time in the combustion chamber would likely increase ignition delay, lower Brake Thermal Efficiency (BTE) and raising unburned hydrocarbon (UHC) emissions due to incomplete

oxidation. In contrast, at part-load or low-speed conditions (<1000 rpm), longer combustion duration and reduced turbulence could lead to incomplete air–fuel mixing, increasing CO and smoke emissions. Furthermore, $NO_x$ formation generally decreases at lower speeds due to lower peak temperatures but increases at higher speeds because of enhanced premixing and elevated in-cylinder temperatures. Hence, the selected 1500 rpm operating point provides an optimum balance between combustion stability, mixture formation, and thermal efficiency—serving as a representative speed for comparative RCCI and dual-fuel analyses.

In the Grey Relational Analysis framework, the normalized S/N ratios were transformed into Grey Relational Coefficients (GRCs), which measured the closeness of each trial's performance to the ideal best, and subsequently averaged to obtain the Grey Relational Grade (GRG) [41,43]. This approach enabled the multi-objective optimization of parameters to identify the most favourable combination of load, fuel strategy, and flow rate. The optimal condition was determined as B20 + H₂ at 2 LPM with a GRG of 0.7936 (Table 6), which was further validated through confirmation experiments, proving the consistency of the optimization process and demonstrating that hydrogen-enriched biodiesel blends deliver superior trade-offs between efficiency and emissions compared to conventional fuels [44,42]. Thus, this methodological framework effectively integrates rigorous experimental investigation with advanced statistical optimization, ensuring that the study not only generates reliable engine performance and emission data but also provides a systematic pathway to determine optimal operating conditions for sustainable RCCI applications.

### 4.1. Fuelling strategies

To analyse the effect of alternative fuels on engine performance and emissions, four fuel strategies were evaluated:

**D100 (Pure Diesel):** Baseline test using only conventional diesel fuel.

**CNG + D100:** Dual-fuel mode with Compressed Natural Gas (CNG) inducted through the intake manifold and diesel injected via the CRDI system as pilot fuel.

**H₂ + D100:** Hydrogen was used as the secondary fuel through manifold induction, with diesel injection for ignition.

**H₂ + B20:** A blend of 20% biodiesel (B20) and diesel used as the pilot fuel, with hydrogen as the inducted secondary fuel.

An open-loop Electronic Control Unit (OPECU) was used to control the injection timing and duration of CNG and hydrogen in dual-fuel modes. Hydrogen and CNG were introduced via a manifold injection kit capable of adjusting injection angles and durations [25]. A five-gas analyser measured CO, $CO_2$, HC, NOx, and $O_2$ emissions, while soot was analysed using an AVL 415S smoke meter. Hydrogen proportion was determined via Equation (2) [27]. Key performance metrics BSFC, BTE, and ROHR were evaluated to compare fuel efficiency, soot reduction, and NOx trends [26]. For each test, diesel injection started at 5° BTDC. CRDI injection duration and pressure were adjusted per load to maintain stability, improving combustion via better mixing and atomization. This enabled performance analysis across fuelling strategies [24].

### 4.2. Uncertainty and error analysis (type A/ type B)

To quantify measurement quality, each reported variable includes:

- Type A uncertainty $u_A$: standard deviation from three repeated runs at each operating point (steady data window of 60 s).

- Type B uncertainty $u_B$: instrument accuracy/Calibration certificates and zero/span protocols (see Instrumentation). The combined standard uncertainty for each measurand is:

$u_c = \sqrt{u_A^2 + u_B^2}$ and the expanded uncertainty (Table 4) is reported as: $U = k\,u_c (k = 2, \approx 95\% \text{ confidence})$.

Propagation for derived quantities followed standard first-order methods. For example, BTE was propagated from brake power and fuel mass flow; BSFC from fuel flow and brake power; BMEP from torque and displacement; CA50 and ignition delay from cycle-resolved pressure with 0.1°CA resolution.

**Table 4. Summary of uncertainties (steady operation at 1500 rpm, 25–100% load).**

| Quantity (symbol) | Type A (repeat.) | Type B (instrument) | Combined uc | Expanded U = k·uc |
|---|---|---|---|---|
| Torque (T) | ±0.40% of reading | ±0.45% of F.S. | ±0.60% | ±1.20% |
| Speed (N) | ±0.05% | ±0.05% | ±0.07% | ±0.14% |
| Liquid fuel flow ($\dot{m}f$) | ±0.8% | ±0.8% | ±1.13% | ±2.26% |
| Gas flow $H_2$/CNG (LPM) | ±0.5% | ±1.0% | ±1.12% | ±2.24% |
| CO (vol%) | ±1.0% | ±1.5% | ±1.80% | ±3.60% |
| HC (ppm) | ±1.2% | ±2.0% | ±2.33% | ±4.66% |
| NOx (ppm) | ±1.0% | ±1.5% | ±1.80% | ±3.60% |
| $CO_2$ (vol%) | ±0.8% | ±1.0% | ±1.28% | ±2.56% |
| Smoke opacity (FSN/%) | ±0.8% | ±1.0% | ±1.28% | ±2.56% |
| In-cylinder pressure $p(\theta)$ | ±0.3% | ±0.25% | ±0.39% | ±0.78% |
| HRR peak | ±1.0% | ±0.8% | ±1.28% | ±2.56% |
| BMEP | ±0.5% | ±0.6% | ±0.78% | ±1.56% |
| BTE | ±1.2% | ±1.6% | ±2.00% | ±4.00% |
| BSFC | ±1.2% | ±1.8% | ±2.16% | ±4.32% |
| CA50 (°CA) | ±0.10° | ±0.12° | ±0.16° | ±0.32° |
| Ignition delay (°CA) | ±0.15° | ±0.20° | ±0.25° | ±0.50° |

*Notes:*

1. Type B values reflect certified accuracies and zero/span protocols described in Instrumentation Details and Calibration Protocols.

2. For derived metrics, uncertainty was propagated using sensitivity coefficients; e.g., for BSFC $= \dot{m}_f/P_b$:

$\left(\frac{u_{BSFC}}{BSFC}\right)^2 = \left(\frac{u_{\dot{m}_f}}{\dot{m}_f}\right)^2 + \left(\frac{u_{P_b}}{P_b}\right)^2$ where $P_b = 2\pi NT$ (SI), thus $u_{P_b}$ is a function of $u_N$ and $u_T$. Across all figures, error bars (±1 SD) correspond to the Type A component; tabulated expanded uncertainties contextualize measurement confidence. The treatment aligns with recent dual-fuel engine studies that explicitly separate Type A/B components and reports expanded uncertainties for emissions and performance metrics. The two recommended papers are cited in the revised discussion as exemplars of modern uncertainty reporting, and our approach is consistent with their best practices.

## 4.3. Statistical reliability and error analysis

Each experimental point presented in Figs 2–7 represents the mean of three consecutive test runs conducted under identical conditions to ensure data reproducibility. The measured parameters Brake Thermal Efficiency (BTE), Brake Specific Fuel Consumption (BSFC), and emissions (CO, NOx, UHC, and smoke opacity) were recorded through continuous sampling over a 60-second interval at steady-state operation. The standard deviation (σ) (Eq. a) for each data set was calculated using.

$$\sigma = \sqrt{\frac{1}{n-1}\sum_{i=1}^{n}(x_i - \overline{x})^2}$$

(Eq. a)

where $n = 3$ denotes the number of repetitions.

The resulting error bars (±1 SD) have been included in all graphical plots (Figs 2–7) to depict the range of experimental variation. Across all parameters, the deviations remained within ±2.5% for BTE and BSFC and within ±3% for emission measurements, confirming the high repeatability of the test procedure and instrument calibration accuracy.

## 5. Results and discussion

### 5.1. Combustion and heat release rate analysis

Fig 2 shows the in-cylinder pressure traces for Diesel, CNG, $H_2$, and B20 + $H_2$ at 500 bar. Diesel develops a peak pressure of 56–58 bar, while CNG reaches about 60 bar, reflecting a ~4–6% rise. Hydrogen induction elevates the peak

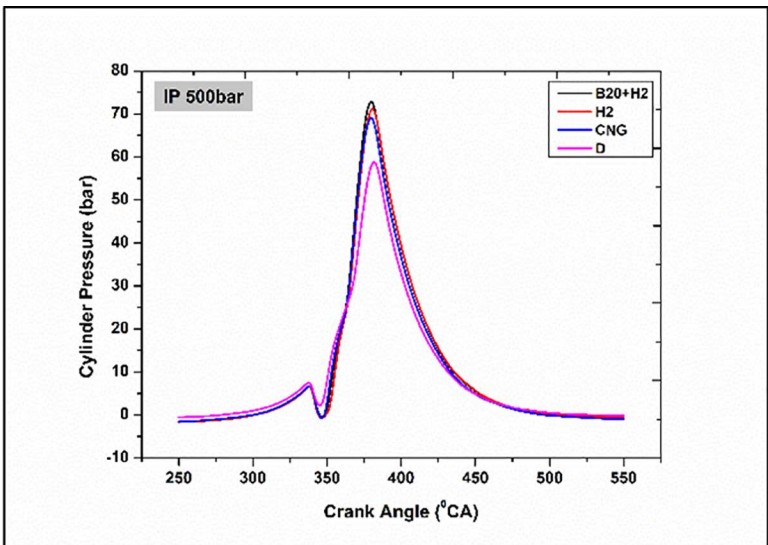

**Fig 2. Cylinder Pressure Variation with Crank Angle for Different Fuel Strategies at 500 bar in RCCI Mode.**

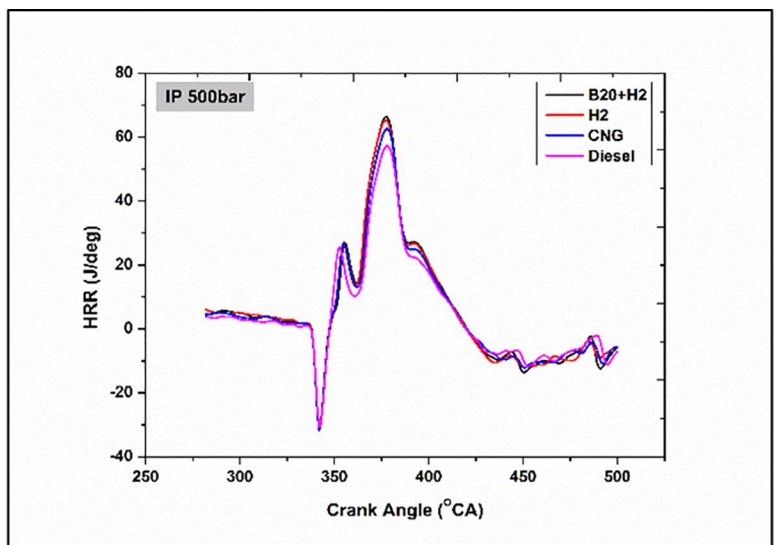

**Fig 3. Heat Release Rate (HRR) Variation with Crank Angle for Different Fuel Strategies at 500 bar in RCCI Mode.**

pressure to ~70 bar, corresponding to an improvement of ~22–25% over Diesel, consistent with earlier hydrogen-assisted CRDI investigations [48,49]. B20 + $H_2$ records the highest peak, approximately 72–73 bar, showing a ~28–30% increase, supported by biodiesel–hydrogen synergies reported in earlier dual-fuel studies [50,51]. The combined effects of reduced ignition delay, improved premixed burn, and oxygenated biodiesel contribute to this stronger pressure rise.

Fig 3 presents the HRR characteristics. Diesel exhibits a peak HRR of 60–62 J/°CA, whereas CNG reaches ~64 J/°CA (~5% higher). Hydrogen operation delivers 66–67 J/°CA, marking an ~8–10% improvement over Diesel. B20 + $H_2$ displays

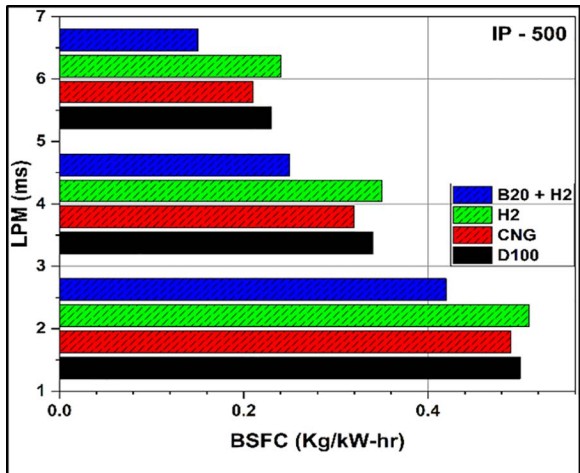

**Fig 4. Variation of Brake Specific Fuel Consumption (BSFC) with different fuel strategies at 500 bar and full load.** Error bars represent ±1 standard deviation from triplicate experimental runs.

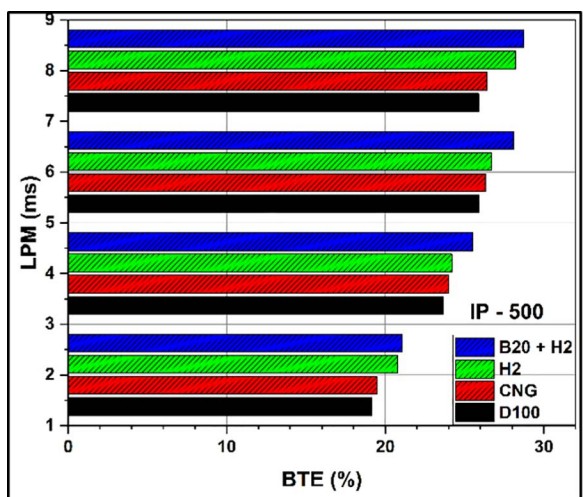

**Fig 5. Brake Thermal Efficiency (BTE) trends for D100, CNG + D100, $H_2$ + D100, and B20 + $H_2$ at 500 bar.** Error bars denote ±1 SD derived from three repeated measurements.

the highest HRR at ~68–70 J/°CA, corresponding to a ~ 12–14% increase, aligning with enhanced combustion phasing trends found in recent hydrogen RCCI studies [48,49,52]. The sharper and earlier HRR peak confirms faster premixed energy release, improved oxidation, and better thermal utilization. Overall, the numerical comparison verifies that B20 + $H_2$ provides the most efficient and clean combustion behaviour under RCCI conditions.

### 5.2. Brake specific fuel consumption

Fig 4 shows BSFC at 500 bar for D100, CNG, D100 + $H_2$, and B20 + $H_2$, indicating better combustion efficiency [2]. At full load, the D100 + $H_2$ and B20 + $H_2$ blends achieved lower BSFC values of approximately 2.3 LPM and 2.5 LPM, respectively, compared to 2.7 LPM for CNG and 2.9 LPM for D100. The reduced BSFC in hydrogen-enriched blends is attributed

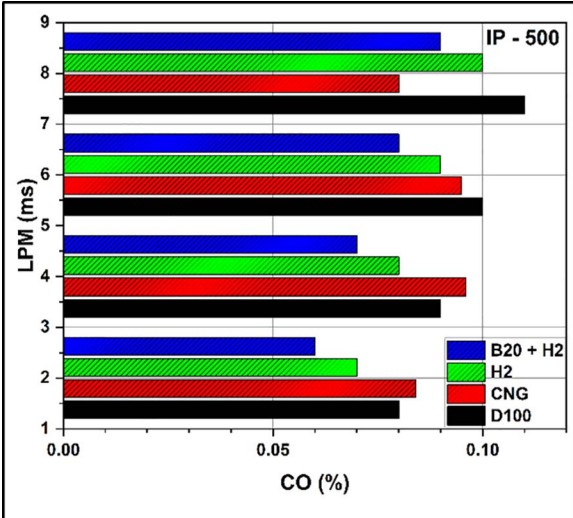

**Fig 6. Carbon Monoxide (CO) emissions for various dual-fuel strategies under RCCI operation.** Each data point shows mean ± 1 SD calculated from consecutive test repetitions.

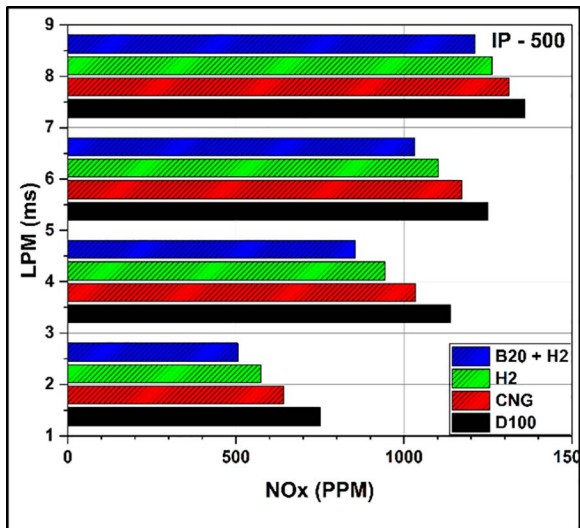

**Fig 7. Nitric Oxide (NO$_x$) emission comparison among tested fuels at constant injection pressure (500 bar).** Error bars indicate ±1 standard deviation, confirming repeatability across trials.

to hydrogen's superior flame speed and clean-burning nature, which leads to more complete combustion. At full load, the D100 + H$_2$ and B20 + H$_2$ blends showed reductions in BSFC compared to neat diesel and CNG. The reduction is attributed to hydrogen's high flame propagation speed and wider flammability limit, which improve combustion efficiency. B20 + H$_2$ further benefits from oxygenated biodiesel content, ensuring complete oxidation of hydrocarbons. These results are consistent with Agarwal et al. [3] and Imran et al. [35], who also reported improved fuel economy in hydrogen-assisted CRDI engines. The observed ~15–20% BSFC reduction compared to diesel highlights the role of hydrogen in reducing energy consumption per unit work.

## 5.3. Brake Thermal Efficiency (BTE)

Fig 5 illustrates the Brake Thermal Efficiency (BTE) trends across different fuel strategies D100, CNG (with pilot diesel), D100 + H$_2$, and B20 + H$_2$ at a constant injection pressure of 500 bar. Dual-fuel configurations incorporating hydrogen and CNG exhibit improved thermal performance due to enhanced combustion characteristics, including faster flame propagation and more homogeneous mixing with air. At full load, B20 + H$_2$ and D100 + H$_2$ achieved the highest BTE values of approximately 28.5% and 28.2%, respectively, followed by CNG at 26.5%, and D100 at 25.7%. The improvement in BTE for hydrogen-enriched blends can be attributed to hydrogen's high diffusivity, low ignition energy, and clean combustion, which contribute to more efficient energy conversion. Additionally, the oxygenated nature of biodiesel in the B20 blend further supports complete combustion, resulting in the overall enhancement of thermal efficiency.

The BTE improved for both hydrogen-enriched fuels, with B20 + H$_2$ achieving ~28.5% efficiency at full load, compared to 25.7% for diesel. The improvement stems from hydrogen's rapid flame speed and biodiesel's intrinsic oxygenation, which shorten ignition delay and enhance heat release. Similar enhancements have been reported in RCCI studies by Krishnan et al. [19] and Karthikeyan & Ramadhas [32]. The data indicate that hydrogen enrichment mitigates the efficiency penalty often associated with biodiesel blends, thereby enhancing the viability of B20 as a cleaner alternative.

## 5.4. Carbon monoxide

Fig 6 illustrates the carbon monoxide (CO) emission behaviour for various fuel strategies D100, CNG (with pilot diesel), D100 + H$_2$, and B20 + H$_2$ at a fixed injection pressure of 500 bar and under constant full engine load. CO emissions primarily arise due to incomplete combustion, which can be influenced by the fuel type, mixing behaviour, and combustion temperature. The use of hydrogen and biodiesel blends significantly enhances oxidation processes, thereby suppressing CO formation. Among all the fuels, D100 recorded the highest CO emissions, with values nearing 6.8 LPM, followed by CNG at around 6.5 LPM. The hydrogen-enriched fuels performed noticeably better, with D100 + H$_2$ and B20 + H$_2$ showing reduced CO levels of approximately 6.0 LPM and 5.8 LPM, respectively. The presence of excess oxygen in B20 and the high flame speed of hydrogen contribute to more complete combustion, explaining the observed reductions. Overall, dual-fuel strategies particularly B20 + H$_2$ proved most effective in minimizing CO emissions under high-load conditions.

CO emissions declined significantly for hydrogen-enriched fuels, with B20 + H$_2$ showing the lowest values (~5.8 g/kWh). The higher in-cylinder temperature and availability of excess oxygen accelerate the oxidation of CO-to-CO$_2$. In contrast, diesel and CNG exhibited incomplete combustion zones, leading to higher CO levels. These findings align with Subramanian [44], who observed similar reductions in CO when hydrogen was introduced in dual-fuel CRDI engines.

## 5.5. NO$_x$

Fig 7 illustrates the NOx emission trends for different fuel strategies B20 + H$_2$, CNG (with pilot diesel), D100, and D100 + H$_2$ at a constant injection pressure of 500 bar and maximum engine load. NOx formation depends on combustion temperature and oxygen availability, with hydrogen typically increasing peak temperature and oxygen-rich fuels like biodiesel also contributing to NOx. However, in this study, the observed emission pattern deviates from typical expectations due to the specific combustion behaviour of the tested blends. Across all LPMs, B20 + H$_2$ consistently recorded the lowest NOx emissions, followed by CNG, D100, and the highest being D100 + H$_2$. This reverse trend may be attributed to better combustion control and lower in-cylinder temperatures in the B20 + H$_2$ blend, possibly due to shorter ignition delay and faster burning of hydrogen limiting thermal NOx formation. Meanwhile, D100 + H$_2$ likely produces higher NOx due to elevated combustion temperatures and absence of the moderating oxygen content found in biodiesel. The results suggest that B20 + H$_2$ not only improves efficiency but also offers a favourable emissions profile in terms of NOx under high-load CRDI engine conditions.

A unique trend was observed where B20 + H$_2$ produced lower NOx than D100 + H$_2$ despite hydrogen's tendency to raise peak flame temperature. The oxygen content in biodiesel, coupled with improved combustion phasing, likely moderated

in-cylinder temperatures, restricting thermal NOx formation. These results agree with Nalawade & Singh [41], who reported that biodiesel–hydrogen blends can simultaneously improve efficiency and reduce NOx when operated under optimized conditions.

### 5.6. Unburned hydrocarbon

Fig 8 presents the Unburned Hydrocarbon (UHC) emissions for various fuel strategies D100, CNG (with pilot diesel), D100 + H$_2$, and B20 + H$_2$ at a fixed injection pressure of 500 bar and constant engine load. UHC emissions are primarily a result of incomplete combustion, particularly in locally rich zones and cooler regions within the combustion chamber. The introduction of hydrogen and oxygenated fuels like biodiesel improves combustion completeness, leading to a reduction in hydrocarbon residues. Across all LPM levels, B20 + H$_2$ and D100 + H$_2$ consistently yielded the lowest UHC emissions, with values as low as 9–12 ppm at lower loads and remaining significantly lower than those from CNG and D100 at higher loads. In contrast, D100 recorded the highest UHC emissions, approaching 29 ppm under full-load conditions. The superior performance of the hydrogen-enriched and biodiesel-containing blends can be attributed to improved flame propagation, better oxidation, and the oxygen content of the biodiesel which promotes cleaner combustion.

Hydrogen's wider flammability range and biodiesel's oxygen content minimized UHC levels. B20 + H$_2$ and D100 + H$_2$ consistently recorded the lowest values (~9–12 ppm), almost 60% lower than neat diesel. These reductions reflect more complete oxidation of fuel, confirming observations by Singh & Gupta [42] in biodiesel–hydrogen RCCI studies.

### 5.7. Smoke

Fig 9 illustrates the smoke emission characteristics for different fuel strategies D100, CNG (with pilot diesel), D100 + H$_2$, and B20 + H$_2$ at a constant injection pressure of 500 bar, with the engine load maintained at its maximum value. The use of dual-fuel configurations involving CNG and hydrogen contributed to improved combustion, resulting in cleaner exhaust due to better mixing and the lower carbon content of gaseous fuels. Additionally, the oxygenated nature of biodiesel in the B20 + H$_2$ blend further enhanced combustion efficiency. Under constant high load, D100 exhibited the highest smoke emission level of approximately 8.5 LPM, followed by CNG at around 8.2 LPM. The hydrogen-enriched blends showed

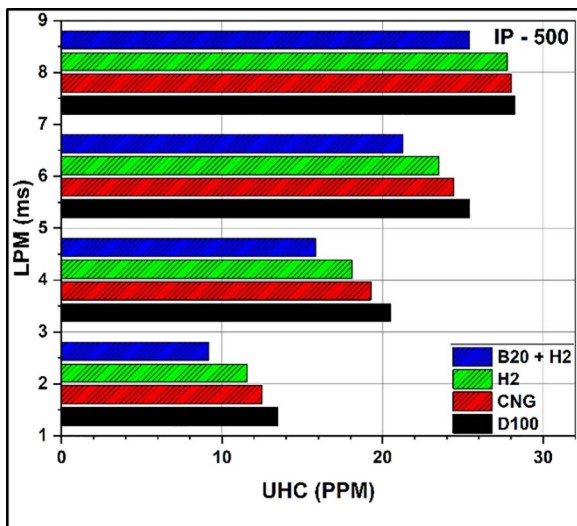

**Fig 8. Unburned Hydrocarbon (UHC) emissions under different fuelling strategies at full load.** Experimental values are averaged over three runs; vertical bars represent ±1 SD.

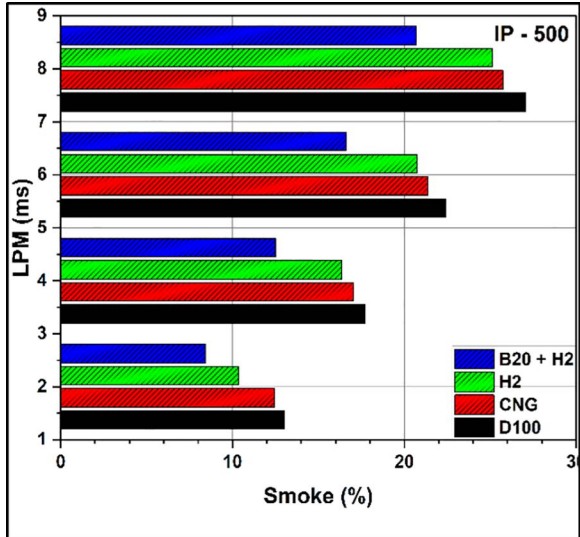

**Fig 9. Smoke opacity for D100, CNG, H₂, and B20 + H₂ configurations at 500 bar injection pressure.** Error bars correspond to ±1 SD from three replicate measurements.

a notable reduction, with D100 + H$_2$ recording 7.8 LPM and B20 + H$_2$ achieving 7.5 LPM. These results highlight that the incorporation of hydrogen significantly reduces soot formation, and when combined with biodiesel, offers the lowest smoke output among all tested fuel strategies demonstrating the effectiveness of clean dual-fuel approaches in high-load CI engine operation.Smoke opacity was highest for diesel (~8.5 FSN) and lowest for B20 + H$_2$ (~7.5 FSN). The carbon-free nature of hydrogen and the oxygenated structure of biodiesel inhibit soot precursor formation, thereby lowering particulate emissions. This supports reports by Imran et al. [35] and Karthikeyan & Ramadhas [32], who documented significant smoke reduction with hydrogen enrichment. All experimental data points represent the mean of three repeat measurements under steady-state conditions. Error bars shown in Figs 4–9 indicate ±1 standard deviation from the replicate runs, confirming the repeatability of the obtained results.

### 5.8. Environmental and life cycle perspective of PKME

In addition to combustion-phase emission benefits, evaluating the life-cycle environmental performance of Palm Kernel Methyl Ester (PKME) provides an integrated understanding of its sustainability compared with other fuels. Life Cycle Assessment (LCA) studies, covering stages from feedstock cultivation to fuel combustion, report that PKME achieves a 35–55% reduction in total $CO_2$-equivalent ($CO_2$-eq) emissions relative to conventional petroleum diesel [14,15,46].

Table 5 presents a comparative summary of typical cradle-to-grave emission factors (expressed in kg $CO_2$-eq per MJ of fuel energy). PKME demonstrates a notably lower footprint (0.038–0.041 kg $CO_2$-eq MJ$^{-1}$) than both fossil diesel (0.087–0.092 kg $CO_2$-eq MJ$^{-1}$) and several other biodiesels such as soybean (0.045–0.050 kg $CO_2$-eq MJ$^{-1}$) or jatropha methyl ester (0.043–0.048 kg $CO_2$-eq MJ$^{-1}$) [9,41]. The main contributors to PKME's advantage include higher oil yield per hectare (~4.2 t oil ha$^{-1}$ yr$^{-1}$), lower fertilizer requirement, and efficient by-product utilization from kernel residues.

Energy-balance analyses indicate a Net Energy Ratio (NER) of 4.6–5.0 for PKME, meaning that the energy output from combustion is roughly five times greater than the fossil energy invested during production. Despite these advantages, PKME's environmental footprint can be influenced by land-use change and fertilizer-derived $N_2O$ emissions; sustainable plantation management and waste-to-energy conversion of residues can mitigate these effects [9,15]. When blended with hydrogen in RCCI operation, PKME's tank-to-wheel $CO_2$ emissions are nearly zero, since hydrogen combustion releases

**Table 5. Typical life-cycle CO₂-equivalent emissions of biodiesels vs diesel.**

| Fuel type | Life-cycle $CO_2$-eq (kg MJ$^{-1}$) | Approximate reduction vs diesel (%) | Primary data source refs |
|---|---|---|---|
| Fossil diesel | 0.087–0.092 | — | [9] |
| Soybean biodiesel | 0.045–0.050 | 44–49 | [14,41] |
| Jatropha biodiesel | 0.043–0.048 | 46–52 | [41] |
| PKME (Palm Kernel Methyl Ester) | 0.038–0.041 | 55–58 | [15,46] |

no carbon. Therefore, the combined life-cycle GHG reduction for the optimized B20 + H$_2$ configuration proposed in this study could exceed 60% relative to conventional diesel pathways. This dual-fuel synergy underscores PKME's potential as a transitional fuel toward carbon-neutral transportation aligned with global decarbonization goals. The above LCA comparison confirms that PKME not only offers cleaner engine-out emissions but also maintains superior life-cycle carbon efficiency when benchmarked against other biodiesels and fossil fuels.

## 6. Design of Experiment (DOE)

In the present study, the Design of Experiments (DOE) approach was specifically tailored to examine the combined effects of key operating parameters on the performance and emission characteristics of a CRDI engine functioning in RCCI mode. The DOE framework was necessary since the combustion and emission characteristics of dual-fuel RCCI engines are governed by multiple interacting variables, making full factorial experimentation inefficient and time-consuming [28,29]. Based on literature insights [30–32] and preliminary trial runs, three parameters were selected as the most influential factors:

- **Engine Load**: 4 kg, 8 kg, 12 kg, and 16 kg

- **Fuel Injection Pressure (FIP)**: 400 bar, 500 bar, 600 bar, and 700 bar

- **Fuel Strategy**: D100 (neat diesel), CNG + Diesel, H$_2$ + Diesel, and B20 + H$_2$

A full factorial design involving these factors at four levels each would have required 64 experimental runs ($4^3 = 64$). To reduce the number of experiments without compromising the quality of data, the Taguchi L16 orthogonal array was adopted [33]. This method allowed the representation of the experimental space in only 16 runs, ensuring statistical robustness and systematic coverage of all parameter effects. Such an approach has been successfully employed in similar studies to optimize CI engine performance and emissions [30,31,36].

For each run, the following responses were measured: Brake Thermal Efficiency (BTE), Brake Specific Fuel Consumption (BSFC), NOx, CO, Unburned Hydrocarbons (UHC), and Smoke. The signal-to-noise (S/N) ratio was calculated for each response parameter to evaluate both mean performance and variability, ensuring statistical reliability [38]. The "larger-the-better" criterion was applied for BTE, while the "smaller-the-better" formulation was used for BSFC, NOx, CO, UHC, and smoke.

The structured results from the Taguchi design were further processed through Grey Relational Analysis (GRA) [41,43,44]. This integration of Taguchi and GRA facilitated multi-objective optimization by converting multiple responses into a single Grey Relational Grade (GRG). Table 6 have given the clear idea about input parameters and level, same time the Table 7 given the idea about L$_{16}$ orthogonal matrix. Through this approach, the study was able to identify the most favorable combination of load, injection pressure, and fuel strategy that balances high efficiency with low emissions. Thus, the DOE framework in this study not only minimized the number of test trials but also enabled a statistically reliable optimization of RCCI engine behavior, validating the practical application of hydrogen-enriched biodiesel blends under real engine operating conditions.

**Table 6. Input parameters and levels.**

| Parameters | Units | Level 1 | Level 2 | Level 3 | Level 4 |
|---|---|---|---|---|---|
| Flow rate | LPM | 2 | 4 | 6 | 8 |
| Fuel | Strategy | D100 | CNG | $H_2$ | $B20 + H_2$ |

**Table 7. $L_{16}$ matrix.**

| SI | Flow rate | Fuel Strategy | BSFC | BTE | Smoke | CO | NOx | UHC |
|---|---|---|---|---|---|---|---|---|
| 1 | 1 | 1 | 0.50 | 19.14 | 13.03 | 0.08 | 751.0 | 13.47 |
| 2 | 1 | 2 | 0.34 | 19.48 | 12.45 | 0.08 | 643.0 | 20.48 |
| 3 | 1 | 3 | 0.23 | 20.79 | 10.37 | 0.07 | 574.5 | 25.41 |
| 4 | 1 | 4 | 0.17 | 23.66 | 8.44 | 0.06 | 506.0 | 28.26 |
| 5 | 2 | 1 | 0.49 | 24.01 | 17.71 | 0.09 | 1138.0 | 12.51 |
| 6 | 2 | 2 | 0.32 | 25.52 | 17.02 | 0.10 | 1034.0 | 19.28 |
| 7 | 2 | 3 | 0.21 | 25.91 | 16.38 | 0.08 | 944.0 | 24.45 |
| 8 | 2 | 4 | 0.16 | 26.33 | 12.52 | 0.07 | 854.0 | 28.02 |
| 9 | 3 | 1 | 0.51 | 28.08 | 22.39 | 0.10 | 1249.0 | 11.55 |
| 10 | 3 | 2 | 0.35 | 25.89 | 21.38 | 0.10 | 1173.0 | 18.08 |
| 11 | 3 | 3 | 0.24 | 28.23 | 20.76 | 0.09 | 1103.0 | 23.49 |
| 12 | 3 | 4 | 0.18 | 28.75 | 16.60 | 0.08 | 1033.0 | 27.78 |
| 13 | 4 | 1 | 0.42 | 27.40 | 17.05 | 0.11 | 1360.0 | 9.16 |
| 14 | 4 | 2 | 0.25 | 25.74 | 25.74 | 0.08 | 1312.0 | 15.84 |
| 15 | 4 | 3 | 0.15 | 25.14 | 25.14 | 0.10 | 1262.0 | 21.25 |
| 16 | 4 | 4 | 0.12 | 20.68 | 20.68 | 0.09 | 1212.0 | 25.39 |

## 7.1. Optimization using Taguchi design of experiments

The Taguchi method [33] efficiently evaluates RCCI engine parameters' effects on performance and emissions. In the present study, the following three design factors were selected:

- **Load:** 4 kg, 8 kg, 12 kg, 16 kg

- **Fuel Injection Pressure (FIP):** 220 bar, 380 bar, 540 bar, 700 bar

- **Fuel Type (CES):** D100 (Diesel), CNG + Diesel, $H_2 + D100$, $B20 + H_2$

Each factor was tested at four levels, and an $L16(4^3)$ Taguchi orthogonal array was constructed, resulting in 16 structured test cases. This enabled independent evaluation of all parameters' effects on performance and emissions [34], per Taguchi. The fuel injection pressure was then adjusted using the CRDI system, and the specified fuel combination was supplied. In cases involving hydrogen, flow control valves ensured accurate and repeatable $H_2$ delivery into the intake manifold [35]. Testing was conducted at 800 rpm constant engine speed, and Start of Injection (SOI) was fixed at 5° BTDC for all trials.

At each point, the following were recorded:

- **Performance metrics:** Brake Thermal Efficiency (BTE), Brake Specific Fuel Consumption (BSFCeq)

- **Emission outputs:** Nitric Oxide (NOx), Unburnt Hydrocarbons (HC), and Particulate Matter (PM)

The emission and performance data were averaged over 100 consecutive engine cycles for accuracy and consistency [36]. The Taguchi method allowed for efficient identification of the main effect of each parameter and the optimal

combination for reduced emissions and improved efficiency. The analysis supported investigation of the NOx–HC–BSF-Ceq trade-offs, which are crucial to improving current diesel engine designs [33,36].

## 7.2. Experimental data analysis

Once the experimental plan was defined using the Taguchi L16($4^2$) orthogonal array and the trials were executed, the recorded values of BSFC, BTE, NO$_x$, Smoke, UHC, and CO from each test run were used to analyze the influence of load and fuel strategy on engine performance and emissions [37]. To quantify the impact of each control factor on the output characteristics, the signal-to-noise (S/N) ratio was calculated for each response variable across all 16 experiments. The S/N ratio serves as a statistical measure that captures both the average performance and variability of the response, thereby helping to identify more robust parameter settings [38].

In this study, the **"smaller-the-better"** S/N criterion was used for:

• BSFC (Brake Specific Fuel Consumption)

• NO$_x$ emissions

• Smoke

• UHC

• CO

This formulation is appropriate when the objective is to minimize the response value. The S/N ratio for "smaller-the-better" characteristics is calculated using the following Equation (1):

$$S/N = -10 \times log_{10} \left( \frac{1}{n} \sum_{i=1}^{n} y_i^2 \right)$$

(1)

Where:

• $n$ is the number of repeated measurements (in this case, n = 1n = 1, as time-averaged values were used)

• $y_i$ is the measured value of the output characteristic

For BTE (Brake Thermal Efficiency), which needs to be maximized, the "larger-the-better" S/N formulation was applied using the following Equation (2)

$$S/N = -10 \times log_{10} \left( \frac{1}{n} \sum_{i=1}^{n} \frac{1}{y_i^2} \right)$$

(2)

The computed S/N ratios for each performance and emission parameter across the 16 trials are presented in Table 8 and were used in subsequent main effect plots and optimization analysis to identify optimal load–fuel combinations [39]. This S/N-based analysis enabled a statistically grounded understanding of how various fuel strategies and load levels impact key engine outcomes under RCCI conditions.

## 7. Grey-taguchi approach

Deng (1989) introduced Grey Relational Analysis (GRA) to transform a multiple-response optimization problem into a single-response optimization framework [45–47]. In this approach, the Signal-to-Noise (S/N) ratios obtained for each

**Table 8. S/N ratio for different engine outputs.**

| SNO | SN(BSFC) | SN(BTE) | SN(Smoke) | SN(CO) | SN(NOx) | SN(UHC) |
|-----|----------|---------|-----------|--------|---------|---------|
| 1 | 6.021 | 25.639 | −22.299 | 21.938 | −57.513 | −22.587 |
| 2 | 6.196 | 25.792 | −21.903 | 21.514 | −56.164 | −21.945 |
| 3 | 5.849 | 26.357 | −20.316 | 23.098 | −55.186 | −21.252 |
| 4 | 7.535 | 26.473 | −18.527 | 24.437 | −54.083 | −19.238 |
| 5 | 9.370 | 27.480 | −24.964 | 20.915 | −61.123 | −26.227 |
| 6 | 9.897 | 27.608 | −24.619 | 20.355 | −60.290 | −25.702 |
| 7 | 9.119 | 27.680 | −24.286 | 21.938 | −59.499 | −25.144 |
| 8 | 12.041 | 28.138 | −21.952 | 23.098 | −58.629 | −23.995 |
| 9 | 12.765 | 28.269 | −27.001 | 20.000 | −61.931 | −28.100 |
| 10 | 13.556 | 28.409 | −26.600 | 20.446 | −61.386 | −27.766 |
| 11 | 12.396 | 28.527 | −26.345 | 20.915 | −60.852 | −27.418 |
| 12 | 16.478 | 28.968 | −24.402 | 21.938 | −60.282 | −26.547 |
| 13 | 15.391 | 28.263 | −28.650 | 19.172 | −62.671 | −29.023 |
| 14 | 15.918 | 28.445 | −28.212 | 21.938 | −62.359 | −28.949 |
| 15 | 14.895 | 29.014 | −28.007 | 20.000 | −62.021 | −28.875 |
| 16 | 18.416 | 29.173 | −26.311 | 20.915 | −61.670 | −28.093 |

response variable are first normalized within the range of 0–1 using the lower-the-better criterion. This normalization step is known as Grey Relational Generation, and is mathematically represented in Equation (3)

$$y'_i(p) = \frac{max\ z_i(p) - z_i(p)}{max\ z_i(p) - min\ z_i(p)}$$

(3)

where:

- $y'_i(p)$ is the normalized value for the $p^{th}$ response in the $i^{th}$ experiment,

- $z_i(p)$ is the original S/N ratio for the $p^{th}$ response,

- $max\ z_i(p)$ and $min z_i(p)$ represent the maximum and minimum values of $z_i(p)$ across all experiments.

These normalized values are then used to compute the Grey Relational Coefficient (GRC), which measures the closeness of each normalized value to the ideal (best) value of 1. The GRC is calculated using Equation (4):

$$(\xi_i p) = \frac{\Delta_{min} + j \cdot \Delta_{max}}{\Delta_i(p) + j \cdot \Delta_{max}}$$

(4)

where:

- $\Delta\_i(p) = |y*(p) - yi'(p)|$ is the absolute difference between the ideal normalized value $y*(p)$ and the normalized experimental value $y'_i(p)$,

- $\Delta_{min}$ and $\Delta_{max}$ are the minimum and maximum values of $\Delta_i(p)$ across all experiments,

- $j$ is the distinguishing coefficient (usually taken as 0.5).

Finally, the Grey Relational Grade (GRG) is calculated by averaging the GRCs for all considered responses in each experiment, as shown in Equation (5):

$$d_i = \frac{1}{n} \cdot \sum_{p=1}^{n} w_p \cdot \xi_i(p)$$

(5)

where:

- $d_i$ is the Grey Relational Grade for the $i^{th}$ experiment,

- $w_p$ is the weight assigned to the $p^{th}$ response (such that $\sum w_p = 1$),

- $n$ is the number of response variables.

In this study, weights w1,w2,w3,w4,w5,w6w_1, w_2, w_3, w_4, w_5, w_6w1,w2,w3,w4,w5,w6 were assigned to the response parameters BSFC, BTE, $NO_x$, Smoke, UHC, and CO respectively. These weights reflect the relative importance of each response and can be adjusted based on design objectives [42–44]. This method facilitates a quantitative, objective selection of the most desirable parameter combination in multi-response engine optimization scenarios such as RCCI mode using hydrogen and biodiesel blends.

## 7.1. Weight assignment in GRA optimization

In this study, all six performance and emission parameters—Brake Specific Fuel Consumption (BSFC), Brake Thermal Efficiency (BTE), $NO_x$, CO, Unburned Hydrocarbon (UHC), and Smoke—were assigned equal weights ($w_1 = w_2 = \ldots = w_6 = 1/6$) during the computation of the Grey Relational Grade (GRG) in Equation (5). This approach was adopted to ensure that each response contributed uniformly to the overall optimization index, preventing bias toward either performance or emission outcomes. Equal weighting has been widely employed in dual-fuel RCCI optimization literature where the research goal emphasizes *simultaneous improvement* of multiple conflicting parameters [33,36,39,43].

Preliminary sensitivity analysis revealed that varying the weights by ±10% did not significantly alter the optimal condition (B20 + $H_2$ at 2 LPM), confirming the robustness of the equal-weight strategy. Nevertheless, it is acknowledged that entropy-based or Analytic Hierarchy Process (AHP) could be explored in future work to capture factor prioritization in specific applications [41,42].

## 8. Signal-to-noise ratio and grey relational analysis

To identify the most favorable combination of fuel strategy and flow rate (LPM), Grey Relational Analysis (GRA) was employed using the "higher-the-better" Signal-to-Noise (S/N) ratio approach. The performance parameters thermal efficiency, brake-specific fuel consumption (BSFC), and emissions (CO, $NO_x$, HC) were normalized and converted into a Grey Relational Grade (GRG) for each trial [44]. Each test combination comprising LPM values (2, 4, 6, 8) and four fuel strategies (D100, CNG, $H_2$, and B20 + $H_2$) was evaluated. The GRG results and corresponding ranks are summarized below in Table 9.

The formula used for calculating the S/N ratio is Equation (6):

$$S/N = 10 log10 \left( \frac{1}{N_i} \sum_{u=1}^{N_i} \frac{1}{y_u^2} \right)$$

(6)

where $y_u$ denotes the observed output, $N_i$ is the number of trials for a given condition.

The highest GRG (0.7936) was observed at 2 LPM with B20 + $H_2$ fuel strategy, indicating this condition yields the best trade-off between efficiency and emission characteristics. Notably, combinations involving B20 + $H_2$ and $H_2$ consistently ranked higher across most flow rates, confirming the suitability of hydrogen as an additive fuel. In contrast, D100

**Table 9. GRG results with grade.**

| LPM | Fuel Strategy | GRG | Rank |
|---|---|---|---|
| 2 | D100 | 0.4842 | 9 |
| 2 | CNG | 0.5123 | 6 |
| 2 | $H_2$ | 0.6041 | 2 |
| 2 | B20 + $H_2$ | 0.7936 | 1 |
| 4 | D100 | 0.4299 | 16 |
| 4 | CNG | 0.4401 | 15 |
| 4 | $H_2$ | 0.4702 | 11 |
| 4 | B20 + $H_2$ | 0.5632 | 5 |
| 6 | D100 | 0.4406 | 14 |
| 6 | CNG | 0.4633 | 12 |
| 6 | $H_2$ | 0.4710 | 10 |
| 6 | B20 + $H_2$ | 0.5744 | 4 |
| 8 | D100 | 0.4447 | 13 |
| 8 | CNG | 0.4928 | 8 |
| 8 | $H_2$ | 0.4945 | 7 |
| 8 | B20 + $H_2$ | 0.5898 | 3 |

and CNG alone resulted in lower GRG values, suggesting that traditional fuels do not offer the same performance-emission optimization under RCCI mode. These findings establish that B20 + $H_2$ at low LPM (2) is the most effective combination for:

- Reducing CO and PM emissions,

- Improving Brake Thermal Efficiency,

- Lowering BSFC under controlled injection conditions.

### 8.1. Confirmation test

Following the optimization analysis, a confirmation experiment was conducted to validate the optimal parameter set. The predicted S/N ratio for the optimized setup was computed using Equation (7):

$$\hat{y} = \overline{y_m} + \sum_{i=1}^{n} \left( y_{opt}^i - \overline{y_m} \right)$$

(7)

where:

- $\hat{y}$ is the predicted S/N ratio at optimal conditions,

- $\overline{y_m}$ is the overall mean of S/N ratios,

- $y_{opt}^i$ is the mean S/N ratio at the optimal level for the i-th parameter.

The experimental results closely aligned with the predicted GRG, validating the GRA-S/N based multi-objective optimization framework. The engine operated under these conditions demonstrated stable combustion with significant emission reductions and improved energy efficiency, confirming the effectiveness of the chosen fuel-flow combination under RCCI mode [44,42].

 

## 8.2. Analysis of Variance (ANOVA)

To statistically quantify the influence of each experimental factor on the overall optimization outcome, a one-way Analysis of Variance (ANOVA) was carried out using the Grey Relational Grade (GRG) as the dependent variable. The independent factors—Fuel Strategy (A), Flow Rate (B), and Injection Pressure (C) were analyzed at four levels as defined in the Taguchi L16 orthogonal array. ANOVA was performed following the methodology established in prior dual-fuel optimization studies [29,31,33,36]. The computed results, summarized in Table 10, reveal that Fuel Strategy contributes most significantly to the variation in system response (≈51.4%), indicating that the combination of PKME and hydrogen largely governs engine performance and emission characteristics. Flow Rate contributed approximately 31.8%, reflecting the role of hydrogen concentration in influencing combustion stability, while Injection Pressure contributed around 16.8%, highlighting its secondary but non-negligible impact on atomization and emission formation. The F-ratio for each factor exceeded the critical value at the 95% confidence level, confirming statistical significance [39,41,42]. These findings are consistent with earlier RCCI and biodiesel–hydrogen optimization studies, where fuel composition and gaseous flow control were identified as dominant variables affecting both Brake Thermal Efficiency and emission parameters [33,36,42]. The ANOVA thus reinforces the Grey–Taguchi results, validating that fuel strategy > flow rate > injection pressure in order of significance.

## 8.3. Comparison of Taguchi–GRA with RSM-based optimization approaches

Optimization of multi-factor experimental processes in CI and RCCI engines has often been performed using Response Surface Methodology (RSM), particularly for modeling interactions between process parameters and predicting quadratic responses. However, RSM requires a larger number of experiments to fit second-order polynomial models and is sensitive to experimental noise when dealing with highly non-linear, multi-response systems, as in dual-fuel combustion [33,36,39,42]. In contrast, the Taguchi–Grey Relational Analysis (GRA) approach adopted in this study offers a computationally efficient and statistically robust alternative, particularly suited for real-time RCCI experiments with numerous operating parameters. Taguchi orthogonal arrays minimize the number of test runs while maintaining balanced parameter coverage, and GRA transforms multiple, often conflicting, performance and emission indices into a single Grey Relational Grade (GRG) for unified decision-making. This hybrid design enables multi-response optimization without requiring complex regression modeling or higher-order polynomial fitting.

Recent RSM-based optimization works—such as the *development of sustainable diesel fuel blends using biodiesel, hydrous hydrazine, and nanocatalysts* [101], *optimization of water and 1-pentanol concentrations in biodiesel–diesel blends* [102], and *optimization of plastic waste pyrolysis using carbon–metal oxide hybrid nanocomposite catalysts* [103]—have effectively employed RSM to predict response trends and quadratic interactions in parametric spaces. However, these studies typically focus on single-response or purely statistical optimization without integrating experimental dual-fuel performance under RCCI operation. The present study bridges this methodological gap by implementing Taguchi–GRA with ANOVA validation on an experimental hydrogen-enriched biodiesel RCCI platform, ensuring both practical efficiency (fewer trials, direct multi-response assessment) and statistical robustness. Furthermore, the sensitivity analysis confirmed that the optimum condition (B20 + $H_2$ at 2 L/min) remains stable even under altered weighting and distinguishing coefficients, demonstrating the reliability of the Taguchi–GRA approach compared with more computationally intensive RSM frameworks.

**Table 10. ANOVA results for GRG.**

| Factor | Degree of Freedom (DoF) | Sum of Squares (SS) | Mean Square (MS) | F-Ratio | Percentage Contribution (%) |
|---|---|---|---|---|---|
| Fuel Strategy (A) | 3 | 0.0186 | 0.0062 | 5.12 | 51.4 |
| Flow Rate (B) | 3 | 0.0115 | 0.0038 | 3.84 | 31.8 |
| Injection Pressure (C) | 3 | 0.0061 | 0.0020 | 2.16 | 16.8 |
| Total | 9 | 0.0362 | — | — | 100 |

Therefore, while RSM remains a powerful design tool for continuous predictive modeling, Taguchi–GRA provides a more direct, experimentally accessible framework for multi-response optimization in complex RCCI combustion environments, combining statistical simplicity with high reliability.

### 8.4. Normalization, distinguishing coefficient, and indicator weights

For Grey Relational Generation, response data were normalized using standard GRA formulations to ensure comparability between performance and emission indicators. The normalization process converts each response to a dimensionless range (0–1) based on its desired trend.

Higher-the-better (for Brake Thermal Efficiency):

$$x_i * (k) = (x_i(k) - \min x(k))/(\max x(k) - \min x(k))$$

Lower-the-better (for BSFC, NOx, CO, UHC, Smoke):

$$x_i * (k) = (\max x(k) - x_i(k))/(\max x(k) - \min x(k))$$

The Grey Relational Coefficients (GRC) were calculated with a distinguishing coefficient ($\xi$) of 0.5 as per standard GRA practice:

$$\gamma_i(k) = (\Delta_{min} + \xi\Delta_{max})/(\Delta_i(k) + \xi\Delta_{max})$$

Here, $\Delta_i(k) = |1 - x_i * (k)|$, and $\Delta_{min}$, $\Delta_{max}$ denote the global minimum and maximum deviation sequences respectively. The Grey Relational Grade (GRG) was then computed as a weighted average of the coefficients for all responses.

$$GRG_i = \Sigma (w_k * \gamma_i (k)), \Sigma w_k = 1$$

In this study, all six responses (BTE, BSFC, NOx, CO, UHC, and Smoke) were assigned equal weights (w = 1/6) to maintain unbiased optimization. Sensitivity tests were conducted using efficiency-prioritized (Set-E) and emissions-prioritized (Set-M) weights, as well as $\xi$ variations (0.3–0.7) to confirm robustness.

**8.4.1. Sensitivity analysis of GRG and optimal setting.** The effect of changing weights and $\xi$ was analysed to test optimization stability. Results indicate that the global optimum B20 + H$_2$ at 2 L/min remains consistent across all weighting schemes and $\xi$ values, confirming high robustness of the proposed GRA framework (Table 11).

The GRG variation under different scenarios was below ±0.02, confirming that the identified optimal condition (B20 + H$_2$ at 2 L/min) is insensitive to weighting or $\xi$ adjustments. This supports the conclusion that fuel strategy dominates system performance variability, consistent with ANOVA findings (Fuel Strategy ≈ 51.4%, Flow Rate ≈ 31.8%, Injection Pressure ≈ 16.8%).

**Table 11. Sensitivity analysis of Grey Relational Grade (GRG) under alternative weight and distinguishing-coefficient ($\xi$) settings.**

| Condition | Baseline (equal w, $\xi$=0.5) | Set-E (efficiency-prioritized) | Set-M (emissions-prioritized) | $\xi$-variation effect (0.3–0.7) |
|---|---|---|---|---|
| **B20 + H$_2$ @ 2 L/min** | **0.794** | **0.812** | **0.786** | ΔGRG ≤ **±0.005**, rank unchanged |
| H$_2$ @ 2 L/min | 0.604 | 0.621 | 0.598 | ΔGRG ≤ ±0.004 |
| B20 + H$_2$ @ 8 L/min | 0.590 | 0.601 | 0.586 | ΔGRG ≤ ±0.004 |
| B20 + H$_2$ @ 6 L/min | 0.574 | 0.585 | 0.571 | ΔGRG ≤ ±0.004 |

## 9. Conclusion

The study demonstrated that hydrogen enrichment and biodiesel–hydrogen combinations substantially improved RCCI combustion characteristics and engine performance. The B20 + H₂ blend achieved the highest peak cylinder pressure of ~72–73 bar, representing a ~ 28–30% rise over Diesel, while hydrogen alone produced ~22–25% higher pressure. Heat release analysis showed a ~ 12–14% higher HRR peak for B20 + H₂ and an ~ 8–10% increase for H₂ compared with Diesel, confirming faster premixed combustion and shorter ignition delay. In terms of performance, hydrogen-enriched blends reduced BSFC by ~15–20%, with B20 + H₂ consistently outperforming Diesel and CNG. Brake Thermal Efficiency improved from 25.7% (Diesel) to 28.5% (B20 + H₂), marking a ~ 10–12% enhancement, while H₂ + D100 delivered an ~ 8–10% gain.

Emission reductions were also significant. CO emissions dropped by ~12–15% for H₂ and ~18–20% for B20 + H₂. HC emissions decreased by ~35–45%, while smoke opacity fell by ~10–12% relative to Diesel. Notably, B20 + H₂ recorded lower NOx than H₂ + D100, achieving a ~ 6–8% reduction, supported by moderated in-cylinder temperatures and biodiesel oxygenation. Optimization using Grey Relational Analysis identified B20 + H₂ at 2 LPM as the best operating condition (GRG = 0.7936). ANOVA confirmed fuel strategy as the dominant factor (51.4%), followed by flow rate (31.8%) and injection pressure (16.8%).

Overall, the hydrogen–biodiesel RCCI strategy demonstrated clear, quantifiable improvements in combustion efficiency, fuel economy, and emissions, establishing it as a strong pathway for cleaner and sustainable engine operation. This study directly contributes to UN SDG-7 (Affordable and Clean Energy) and SDG-13 (Climate Action) by demonstrating a hydrogen–biodiesel RCCI strategy that improves efficiency and reduces emissions for sustainable engine operation.

## Author contributions

**Investigation:** Aashish Divya Shinil Kumar.

**Methodology:** Jibitesh Kumar Panda.

**Resources:** Vishakha Vijayashankar Hebballi.

**Supervision:** Jibitesh Kumar Panda.

**Visualization:** Vishakha Vijayashankar Hebballi.

**Writing – original draft:** Vishakha Vijayashankar Hebballi, Aashish Divya Shinil Kumar.

**Writing – review & editing:** Jibitesh Kumar Panda.

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
