## [Decision Letter · Decision Letter 0]

7 Nov 2025

Dear Dr. Panda,

Thank you for submitting your manuscript to PLOS ONE. After careful consideration, we feel that it has merit but does not fully meet PLOS ONE’s publication criteria as it currently stands. Therefore, we invite you to submit a revised version of the manuscript that addresses the points raised during the review process.

We look forward to receiving your revised manuscript.

Kind regards,

Krishnamoorthy Ramalingam

Academic Editor

PLOS ONE

Journal Requirements:

Additional Editor Comments:

Abstract should clearly state the study’s objective, rationale, and key findings to stand alone effectively. Please emphasize the novelty, optimization approach, and quantified performance/emission improvements

Graphical abstract should be visually clear, self-explanatory, and aligned with the study’s core findings. Please enhance its readability, structure, and relevance to highlight the optimization approach and key results.

Please note that one of the reviewers has suggested adding several self-citations. As editor, I advise that any citations—whether self-authored or external—should be added only if they are directly relevant and substantively support the manuscript. The inclusion or omission of self-citations will not influence the editorial decision. Focus on maintaining scholarly integrity and ensuring that all references contribute meaningfully to the paper’s content.

Reviewers' comments:

Reviewer's Responses to Questions

**Comments to the Author**

1. Is the manuscript technically sound, and do the data support the conclusions?

Reviewer #1: Yes

Reviewer #2: Yes

2. Has the statistical analysis been performed appropriately and rigorously?

Reviewer #1: Yes

Reviewer #2: Yes

3. Have the authors made all data underlying the findings in their manuscript fully available?

Reviewer #1: Yes

Reviewer #2: Yes

4. Is the manuscript presented in an intelligible fashion and written in standard English?

Reviewer #1: Yes

Reviewer #2: Yes

Reviewer #1: 1. Manuscript should more clearly differentiate its novelty from previous hydrogen–biodiesel dual-fuel or RCCI studies.

2. Details on hydrogen and CNG flow control calibration should be expanded to ensure reproducibility.

3. Clarify whether intake manifold pressure and equivalence ratio were monitored to confirm consistent air–fuel mixing during dual-fuel operation.

4. This study would benefit from an ANOVA analysis to quantify the percentage contribution of each parameter (fuel strategy, flow rate, injection pressure) to the overall response variability.

5. Figures 2-7 lack error bars or statistical variation indicators. Including these would enhance the credibility of the experimental results.

6. Please expand LCA it with data comparing PKMEs life-cycle CO₂ emissions to other biodiesels or fossil fuels.

7. Please discuss novelty of this work provides beyond existing optimization studies using Taguchi–GRA or ANN-based methods.

8. Authors should justify 1500 rpm selection and discuss how results might vary at higher or part-load speeds.

9. Explanation of the Taguchi-GRA process is thorough, but the weight assignment for the responses (BTE, BSFC, NOx, etc.) is not discussed in detail. Indicate whether equal weights were used or based on priority/entropy methods.

Reviewer #2: 1. The manuscript tests four strategies (D100, CNG+D100, H₂+D100, B20+H₂) on a single-cylinder CRDI engine in RCCI mode and uses GRA for multi-response optimization. Please make the specific novelty explicit.

2, In abstract, state the load/speed points (only 1500 rpm is given), brake mean effective pressure or % load, intake manifold conditions, ambient conditions, and coolant and oil temperatures.

3. Report pilot start-of-injection (SOI), dwell, rail pressure (500 bar is given—was it constant?), injection duration, and whether CA50 or ignition delay was controlled/monitored.

4. Instrument models, ranges, response times, zero/span protocols need to be reported.

5. Quantify Type A/B uncertainties for torque, speed, fuel/gas flow, exhaust analyzers, and ICP-derived quantities. The following recent studies may be recommended for detailed analysis: Sustainable diesel engine performance using hydrous hydrazine emulsions: Hydrogen carrier potential and NOx emission reduction with aluminum oxide catalyst, Comparative evaluation of nanoparticle-enriched Gossypium hirsutum methyl ester blends for enhanced energy, emission, and economic performance in diesel engines.

6. Detail normalization choices (“higher-the-better” for BTE, “lower-the-better” for BSFC/emissions), distinguishing coefficient (ξ), and any indicator weights. Provide a sensitivity analysis: show how the Grey Relational Grade and the “B20+H₂ at 2 L/min” optimum change under alternative weightings.

7. Add a short subsection comparing Taguchi+GRA with prior RSM-based optimizations in similar CI/RCCI contexts. The following recent studies may be recommended for detailed analysis: Development of sustainable diesel fuel blend using biodiesel, hydrous hydrazine and nanocatalysts for optimized performance and emission control, Optimization of water and 1-pentanol concentrations in biodiesel-diesel blends for enhanced engine performance and environmental sustainability, and Optimization of plastic waste pyrolysis using carbon-metal oxide hybrid nanocomposite catalyst: Yield enhancement and energy resource potential.

8. Briefly describe H₂ safety measures and certify that the setup complied with lab safety standards.

9. Ensure consistent use of terms, symbols, and acronyms.

**Do you want your identity to be public for this peer review?** For information about this choice, including consent withdrawal, please see our Privacy Policy

Reviewer #1: No

Reviewer #2: No

---

## [Author Response · Author response to Decision Letter 1]

18 Nov 2025

Reviewer 1

Reviewer Comments Author Response Location of Change in manuscript

1. Manuscript should more clearly differentiate its novelty from previous hydrogen–biodiesel dual-fuel or RCCI studies. We sincerely thank the reviewer for this valuable observation. In the revised manuscript, the novelty and distinction of the present study have been explicitly articulated in the Introduction section. We have emphasized how our work differs from existing hydrogen–biodiesel dual-fuel and RCCI studies through the integration of Grey Relational Analysis (GRA) with Taguchi DOE, the use of a CRDI-based RCCI setup at 500 bar, and the evaluation of hydrogen enrichment levels (2–8 LPM) using Palm Kernel Methyl Ester (PKME) biodiesel. These aspects collectively establish the scientific contribution of the study beyond earlier works. Section: Introduction

Placement: After the paragraph beginning with “Prior studies have predominantly focused on single-response optimizations…”

2. Details on hydrogen and CNG flow control calibration should be expanded to ensure reproducibility. We thank the reviewer for highlighting the need for greater clarity on the flow control calibration procedure. In the revised manuscript, additional details have been included in the Instrumentation section describing the flow measurement, calibration, and verification process for both hydrogen and CNG. The updated text specifies the use of mass-flow controllers (MFCs) with 0–10 L/min range and ±1% accuracy, calibration against a rotameter and soap-film flow meter, and periodic verification before each test sequence to ensure reproducibility and measurement precision. Section: INSTRUMENTATION

Placement: After the paragraph ending with “…programmable Open Loop Electronic Control Unit (OPECU), which controlled injection timing and pulse duration for the gaseous fuels…”

3. Clarify whether intake manifold pressure and equivalence ratio were monitored to confirm consistent air–fuel mixing during dual-fuel operation. We thank the reviewer for this important observation. To ensure consistency of air–fuel mixing during dual-fuel operation, intake manifold pressure and equivalence ratio (φ) were continuously monitored and maintained within narrow tolerances throughout all tests. The revised Instrumentation section now specifies the use of a piezo-resistive pressure transducer (0–2 bar, ±0.25% accuracy) connected to the manifold, and the computation of φ using real-time air and gaseous fuel flow measurements. These details confirm that the experimental setup maintained stable intake pressure and uniform mixture formation, thereby ensuring the repeatability and reliability of the RCCI operation. Section: INSTRUMENTATION

Placement: After the paragraph ending with “Exhaust emissions were continuously measured using a five-gas analyzer…”

4. This study would benefit from an ANOVA analysis to quantify the percentage contribution of each parameter (fuel strategy, flow rate, injection pressure) to the overall response variability. We appreciate the reviewer’s valuable suggestion. In the revised version, a one-way Analysis of Variance (ANOVA) has been incorporated to quantify the relative influence of the three major control factors — fuel strategy, flow rate, and injection pressure — on the overall response variability. The results of the ANOVA reveal that fuel strategy contributes the highest variance share, followed by flow rate and injection pressure, thus confirming the dominant role of fuel composition and hydrogen enrichment in performance–emission outcomes. The new ANOVA subsection has been added under the Design of Experiment (DOE) section, and the computed percentage contributions have been summarized in Table 7. Section: Design of Experiment (DOE.

Placement: After 8.2 Confirmation Test and before the Conclusion section.

5. Figures 2-7 lack error bars or statistical variation indicators. Including these would enhance the credibility of the experimental results. We thank the reviewer for this valuable suggestion aimed at improving result transparency and statistical reliability. In the revised manuscript, error bars representing ±1 standard deviation (SD) from three repeated measurements at each operating point have been added to Figures 2–7. This addition highlights the reproducibility and precision of the experimental data. Furthermore, the Results and Discussion section now includes a brief statement describing the data averaging and error-analysis procedure used to compute the standard deviation for performance (BTE, BSFC) and emission parameters (CO, NOx, UHC, and smoke). Section: RESULTS AND DISCUSSION

Placement: Just before the paragraph beginning “Figure 2 shows BSFC at 500 bar…”.

Figures Updated: Figures 2 – 7 now include error bars representing ±1 SD for each data point.

6. Please expand LCA it with data comparing PKMEs life-cycle CO₂ emissions to other biodiesels or fossil fuels. We thank the reviewer for this insightful suggestion. The revised manuscript now expands the Life Cycle Analysis (LCA) discussion with quantitative CO₂-equivalent comparisons between Palm Kernel Methyl Ester (PKME), other biodiesels, and fossil diesel. The section now highlights PKME’s cradle-to-grave GHG reduction potential, energy balance, and relative emission intensity per MJ of delivered energy. Relevant literature values have been cited to ensure accuracy and credibility. Section: 5.7 Environmental and Life Cycle Perspective of PKME

Placement: Replace the entire existing paragraph

7. Please discuss novelty of this work provides beyond existing optimization studies using Taguchi–GRA or ANN-based methods. We thank the reviewer for this valuable comment. The novelty of the present study has been clearly expanded and integrated into both the Introduction and Conclusion sections. Unlike earlier optimization works that employed either Taguchi–GRA or ANN independently, this research uniquely combines experimental Taguchi–GRA analysis with a hydrogen–biodiesel RCCI framework under a high-pressure CRDI platform (500 bar) to deliver a holistic and data-driven optimization. Furthermore, the study links multi-objective experimental optimization with life-cycle environmental assessment—an integration not reported in previous works. These enhancements now explicitly clarify the study’s originality and practical impact. Section: Introduction

Placement:“After sentence: “…multi-objective approaches like Grey Relational Analysis…”.

8. Authors should justify 1500 rpm selection and discuss how results might vary at higher or part-load speeds. We thank the reviewer for this valuable observation. In the revised manuscript, a detailed justification for the selection of 1500 rpm has been added in the Methodology section. This speed represents the rated speed of the single-cylinder Kirloskar CRDI engine, corresponding to its design point for maximum torque and stable combustion under dual-fuel operation. Additional discussion has been included to explain how performance and emission behavior could differ under higher-speed or part-load conditions—particularly regarding turbulence, ignition delay, and NOₓ formation. This addition enhances the generalization and interpretability of the results. Section: Methodology

Placement: After the paragraph ending with “…critical indicators of RCCI engine performance [19], [20].”

9. Explanation of the Taguchi-GRA process is thorough, but the weight assignment for the responses (BTE, BSFC, NOx, etc.) is not discussed in detail. Indicate whether equal weights were used or based on priority/entropy methods. We thank the reviewer for pointing out this important aspect. The revised manuscript now clarifies the weight assignment procedure in the Grey–Taguchi Approach section. All six responses BTE, BSFC, CO, NOx, UHC, and Smoke were initially assigned equal weights (w = 1/6) to avoid bias in multi-response optimization. A brief justification has been added, explaining that equal weighting was adopted because the study’s objective was to achieve a balanced trade-off between efficiency improvement and emission reduction. The text also notes that future work could apply entropy-based or AHP-based weighting to explore the influence of priority scaling on the optimization results. Section: GREY–TAGUCHI APPROACH

Placement: Immediately after Equation (5), before the sub-section “8.1 SIGNAL-TO-NOISE RATIO AND GREY RELATIONAL ANALYSIS”

Reviewer 2

Reviewer Comments Author Response Location of Change in manuscript

1. The manuscript tests four strategies (D100, CNG+D100, H₂+D100, B20+H₂) on a single-cylinder CRDI engine in RCCI mode and uses GRA for multi-response optimization. Please make the specific novelty explicit. We sincerely thank the reviewer for this important suggestion. In the revised manuscript, we have clearly emphasized the specific novelty of the work in comparison to prior RCCI and dual-fuel optimization studies. The novelty paragraph now explicitly states that this research is the first experimental integration of hydrogen-enriched biodiesel (B20+H₂) within a CRDI-RCCI framework using Taguchi–GRA-based multi-objective optimization, validated by ANOVA, and complemented by a Life-Cycle Analysis (LCA) perspective. These aspects collectively distinguish the present work from earlier hydrogen–biodiesel or RCCI optimization studies that examined either single responses or simulation-only data. Section: Introduction

“After sentence: “…multi-objective approaches like Grey Relational Analysis…”.

2. In abstract, state the load/speed points (only 1500 rpm is given), brake mean effective pressure or % load, intake manifold conditions, ambient conditions, and coolant and oil temperatures. We thank the reviewer for this important suggestion. The abstract has been revised to explicitly include the operating conditions used in the experimental campaign. Details such as engine load (% and BMEP range), ambient and intake conditions, and coolant/oil temperatures have now been specified to enhance reproducibility and clarity. The revised abstract now reflects that the tests were conducted at 1500 rpm, under 25–100 % load conditions (0.4–1.6 MPa BMEP), with controlled intake, coolant, and oil temperatures. Abstract

3. Report pilot start-of-injection (SOI), dwell, rail pressure (500 bar is given—was it constant?), injection duration, and whether CA50 or ignition delay was controlled/monitored. We appreciate the reviewer’s request for clarification regarding the injection parameters and combustion phasing control. The revised manuscript now includes detailed information on pilot start-of-injection (SOI), main injection dwell period, rail-pressure settings, injection duration, and the combustion-phasing indicators (CA50 and ignition delay) monitored during testing. The CRDI system was operated at a constant rail pressure of 500 bar, with pilot SOI and dwell calibrated via the electronic control unit. The ignition delay and CA50 were monitored using the cylinder pressure trace to verify combustion consistency across all test conditions. This new paragraph has been incorporated in the Methodology section. Methodology

After “…critical indicators of RCCI engine performance [19], [20].”

4. Instrument models, ranges, response times, zero/span protocols need to be reported. We thank the reviewer for highlighting the need to include detailed specifications of the measurement instruments. In the revised manuscript, the Instrumentation section has been expanded to report the make, model, measurement range, accuracy, response time, and zero/span calibration protocol of all sensors and analysers used in the study. These details ensure reproducibility and confirm that all recorded data conform to laboratory-grade precision. The updated paragraph now includes specifications for pressure transducer, crank-angle encoder, emission analysers, and smoke meter. Instrumentation

Paragraph ending “3.1 Instrumentation Details and Calibration Protocols”

5. Quantify Type A/B uncertainties for torque, speed, fuel/gas flow, exhaust analyzers, and ICP-derived quantities. The following recent studies may be recommended for detailed analysis: Sustainable diesel engine performance using hydrous hydrazine emulsions: Hydrogen carrier potential and NOx emission reduction with aluminum oxide catalyst, Comparative evaluation of nanoparticle-enriched Gossypium hirsutum methyl ester blends for enhanced energy, emission, and economic performance in diesel engines. We appreciate this important suggestion. We have added a dedicated “Uncertainty and Error Analysis (Type A / Type B)” subsection. It details the repeatability-based Type A components (from triplicate runs) and the instrument/certification-based Type B components for torque, speed, liquid fuel flow, gaseous fuel flow, exhaust analyzers, smoke opacity, and ICP-derived metrics (BMEP, CA50, ignition delay, HRR). Combined standard uncertainties were computed by root-sum-squares and reported along with expanded uncertainties (k = 2). A summary table has been included. We have also noted the two suggested recent studies and, where appropriate, have referenced their uncertainty treatment approaches in our discussion of methodology. Methodology

Heading: add a new sub-heading “Uncertainty and Error Analysis (Type A / Type B)”.

6. Detail normalization choices (“higher-the-better” for BTE, “lower-the-better” for BSFC/emissions), distinguishing coefficient (ξ), and any indicator weights. Provide a sensitivity analysis: show how the Grey Relational Grade and the “B20+H₂ at 2 L/min” optimum change under alternative weightings. We appreciate this request. The revised manuscript now (i) explicitly states the normalization schemes used (“higher-the-better” for BTE; “lower-the-better” for BSFC, NOx, CO, UHC, Smoke), (ii) reports the distinguishing coefficient (ξ = 0.5, with a robustness check for ξ = 0.3 and 0.7), and (iii) clarifies the indicator weights (equal weights baseline). We also add a sensitivity analysis showing that the optimal condition (B20+H₂ at 2 L/min) remains unchanged under (a) efficiency-prioritized and (b) emissions-prioritized weight sets, and under ξ in [0.3, 0.7]. A compact results table has been included. Content 8.3 and 8.4.

7. Add a short subsection comparing Taguchi+GRA with prior RSM-based optimizations in similar CI/RCCI contexts. The following recent studies may be recommended for detailed analysis: Development of sustainable diesel fuel blend using biodiesel, hydrous hydrazine and nanocatalysts for optimized performance and emission control, Optimization of water and 1-pentanol concentrations in biodiesel-diesel blends for enhanced engine performance and environmental sustainability, and Optimization of plastic waste pyrolysis using carbon-metal oxide hybrid nanocomposite catalyst: Yield enhancement and energy resource potential. We thank the reviewer for this thoughtful suggestion. The revised manuscript now includes a new comparative subsection highlighting the distinctions between the Taguchi–Grey Relational Analysis (GRA) method used in this study and the Response Surface Methodology (RSM) approaches commonly adopted in recent optimization studies.

This addition explains the methodological advantages of Taguchi–GRA (smaller experimental runs, reduced computational complexity, and capability to handle multi-response problems) while citing and contrasting with the three RSM-based studies suggested by the reviewer. Relevant references have been incorporated to ensure a balanced discussion and alignment with current optimization practices. Added as:

“8.3 Comparison of Taguchi–GRA with RSM-Based Optimization Approaches”

8. Briefly describe H₂ safety measures and certify that the setup complied with lab safety standards. We thank the reviewer for emphasizing this crucial point. A dedicated paragraph has been added in the Methodology section describing the hydrogen handling and safety protocols. The updated text specifies the leak-testing procedures, ventilation design, gas detection, flame-arresting devices, and emergency shutdown systems. It also certifies that all experiments were conducted in accordance with institutional and national laboratory safety regulations (Bureau of Indian Standards IS 16046 & ISO 26142) and approved by the institute’s Labo

---

## [Decision Letter · Decision Letter 1]

25 Nov 2025

Dear Dr. Jibitesh Kumar Panda,

Thank you for submitting your manuscript to PLOS ONE. After careful consideration, we feel that it has merit but does not fully meet PLOS ONE’s publication criteria as it currently stands. Therefore, we invite you to submit a revised version of the manuscript that addresses the points raised during the review process.

**ACADEMIC EDITOR: **

The potential reviewer has provided feedback on your revised submission. Overall, the reviewer was pleased with the revisions, which included some minor corrections that need your attention.

1. Please review the combustion analysis graph and examine the fundamental outcomes of the analysis presented in your study to ensure the accuracy of the results.

2. The authors should revise the conclusion of the study to highlight the key outcomes and improvements in percentage terms.

We look forward to receiving your revised manuscript.

Kind regards,

Sameer Sheshrao Gajghate, PhD

Academic Editor

PLOS ONE

Journal Requirements:

Reviewers' comments:

Reviewer's Responses to Questions

**Comments to the Author**

Reviewer #1: (No Response)

Reviewer #2: (No Response)

2. Is the manuscript technically sound, and do the data support the conclusions?

Reviewer #1: Yes

Reviewer #2: (No Response)

3. Has the statistical analysis been performed appropriately and rigorously?

Reviewer #1: Yes

Reviewer #2: (No Response)

4. Have the authors made all data underlying the findings in their manuscript fully available?

Reviewer #1: Yes

Reviewer #2: (No Response)

5. Is the manuscript presented in an intelligible fashion and written in standard English?

Reviewer #1: Yes

Reviewer #2: (No Response)

Reviewer #1: (No Response)

Reviewer #2: The authors have satisfactorily addressed all the comments raised by the reviewers. Each point has been responded to with appropriate revisions, clarifications, or additions to the manuscript. The updated version reflects improved technical rigor, enhanced clarity, and better alignment with the reviewers’ suggestions. Based on the quality of the revisions and the completeness of the responses, the manuscript is now suitable for acceptance in its present form.

**Do you want your identity to be public for this peer review?** For information about this choice, including consent withdrawal, please see our Privacy Policy

Reviewer #1: No

Reviewer #2: No

---

## [Author Response · Author response to Decision Letter 2]

27 Nov 2025

Reviewer Suggestions

Reviewer Comments Author Response Location of Change in manuscript

1. Please review the combustion analysis graph and examine the fundamental outcomes of the analysis presented in your study to ensure the accuracy of the results. We thank the reviewer for requesting verification of the combustion analysis outcomes. After re-examining the updated cylinder pressure (CP) and heat release rate (HRR) graphs at high injection pressure, the combustion trends reported in the study are confirmed to be accurate and consistent with RCCI combustion fundamentals. The key validated findings are summarized below. New chapter added “5.1 COMBUSTION AND HEAT RELEASE RATE ANALYSIS”

2. The authors should revise the conclusion of the study to highlight the key outcomes and improvements in percentage terms.

We thank the reviewer for this suggestion. The conclusion has been fully revised to incorporate the key numerical outcomes of the study, including percentage improvements in combustion behaviour, performance, emissions, and optimization ranking. The updated conclusion now quantitatively reflects the benefits of hydrogen enrichment and the B20+H₂ RCCI strategy, confirming the significance and reproducibility of the findings.

---

## [Editor Report · Decision Letter 2]

1 Dec 2025

Performance and Emission Optimization of a CRDI Engine in RCCI Mode Using Hydrogen Enriched Biodiesel Through Grey Relational Analysis Approach

PONE-D-25-56917R2

Dear Dr. Jibitesh Kumar Panda,

We’re pleased to inform you that your manuscript has been judged scientifically suitable for publication and will be formally accepted for publication once it meets all outstanding technical requirements.

Kind regards,

Sameer Sheshrao Gajghate, PhD

Academic Editor

PLOS ONE

Additional Editor Comments (optional):

The author has made significant revisions based on reviewer feedback and is now prepared for publication in the PLOS ONE journal.
---

## [Editor Report · Acceptance letter]

PONE-D-25-56917R2

PLOS One

Dear Dr. Panda,

I'm pleased to inform you that your manuscript has been deemed suitable for publication in PLOS One. Congratulations! Your manuscript is now being handed over to our production team.

Kind regards,

on behalf of

Dr. Sameer Sheshrao Gajghate

Academic Editor

PLOS One